# Beyond silos: Drivers and barriers to intersectoral collaboration in zoonotic disease surveillance and response in the Greater Accra Metropolitan Area, Ghana

**Joannishka K. Dsani**[1,2*], **Sherry Ama Mawuko Johnson**[3], **Sandul Yasobant**[1,2,4,5], **Walter Bruchhausen**[1,2,6]

1 One Health Graduate School, Center for Development Research (ZEF), University of Bonn, Bonn, Germany, 2 Section Global Health, Institute for Hygiene and Public Health (IHPH), University Hospital Bonn, Bonn, Germany, 3 School of Veterinary Medicine, College of Basic and Applied Sciences, University of Ghana, Legon-Accra, Ghana, 4 Centre for One Health Education, Research and Development (COHERD), Indian Institute of Public Health Gandhinagar (IIPHG), Gandhinagar, Gujarat, India, 5 Department of Public Health Science, Indian Institute of Public Health Gandhinagar (IIPHG), Gandhinagar, Gujarat, India, 6 German-West African Center for Global Health and Pandemic Prevention (G-WAC), Bonn, Germany

* joannishkad@gmail.com

## Abstract

The One Health approach has gained global traction as a strategy to combat zoonotic diseases, which account for 60–75% of emerging infectious diseases. While effective surveillance requires intersectoral collaboration, challenges such as fragmented systems, resource constraints, and weak coordination hinder efforts, particularly in low- and middle-income countries like Ghana. This qualitative study explores the factors influencing collaboration in zoonotic disease surveillance and response at the operational level, providing insights to strengthen intersectoral collaboration and improve public health outcomes across human, animal and wildlife sectors. Using reflexive thematic analysis, we analyzed interviews with 66 professionals from the human, animal, and wildlife health sectors, all directly involved in zoonotic disease surveillance and response. The findings reveal individual factors (interpersonal relationships, personal initiative, motivations, professional hierarchy, shared interests) and structural factors (financial resources, workforce availability, the governance and organization of surveillance and response systems, institutional visibility, knowledge systems) that shape collaboration dynamics. Additionally, positive outcomes from past collaborations created reinforcing cycles that influenced future engagement. Participants shared expectations of improved efficiency, strengthened disease surveillance, and enhanced resource pooling from future collaboration. Despite the global push for intersectoral collaboration, operational-level challenges persist. While grounded in a Ghanaian context, the types of factors identified likely resonate across resource-limited settings, though their specific manifestations and relative importance may vary by context. These findings underscore the broader need for stronger

**Data availability statement:** Data from this study are not publicly available due to ethical and confidentiality restrictions, as the qualitative interview data contain potentially identifiable information. However, sufficient anonymised excerpts supporting the study's findings are included within the paper and its Supporting information files. The underlying data are available from the One Health Graduate School, Center for Development Research (ZEF), University of Bonn, Bonn, Germany, for researchers who meet the criteria for access to confidential data. Requests may be directed to the Coordinator, Forschungskolleg 'One Health', Center for Development Research (ZEF), Genscherallee 3, 53113 Bonn, Germany. Email: health@uni-bonn.de. Data access is managed through this institutional contact to ensure appropriate oversight and long-term availability.

**Funding:** JD and WB received the awards. The main project funding was provided for by the Ministry of Culture and Science of North Rhine-Westphalia, Germany through the grant Forschungskolleg 'One Health and Urban Transformation' (https://www.zef.de/onehealth.html). A travel grant was awarded by the One Health Soulsby Foundation Fellowship, United Kingdom (https://soulsbyfoundation.org). The publication of this work was supported by the Open Access Publication Fund of the University of Bonn. The funders had no role in the study design, data collection, and analysis, decision to publish, or preparation of the manuscript.

**Competing interests:** The authors have declared that no competing interests exist.

governance, equitable partnerships, and realistic alignment of stakeholder expectations to foster sustainable One Health collaboration and enhance zoonotic disease surveillance globally.

## Introduction

Zoonotic diseases account for 60−75% of emerging infectious diseases, posing a significant health threat [1–3]. Outbreaks of COVID-19, Highly Pathogenic Avian Influenza (HPAI), Ebola, and Marburg illustrate how rapidly these diseases can spread, disrupting economies, agriculture, and food production [4,5]. The risk of zoonotic disease transmission is particularly high in low- and middle-income countries (LMICs), where frequent human-animal interactions facilitate zoonotic spillover, while weak surveillance systems and limited public health infrastructure impede effective detection and response [6–8]. Over the past two decades, zoonotic diseases have caused approximately 30,000 human infections in sub-Saharan Africa [9]. Climate change is expected to exacerbate the spread of zoonotic diseases, altering their epidemiology, and increasing the severity of vector-borne, waterborne, foodborne, and airborne infections [10,11].

The One Health approach, which recognizes that human, animal, and environmental health are interconnected and require integrated actions, has gained global momentum as a strategy to combat zoonotic diseases [12–14]. Effective disease surveillance requires intersectoral collaboration; however, governance challenges, fragmented systems, resource constraints and poor communication often hinder these collaborative efforts [12–14]. In LMICs like Ghana, translating One Health principles into practical, sustained collaboration at the operational level remains a major challenge due to limited guidance and weak intersectoral coordination [13].

While research in other LMICs highlight the importance of institutional coordination and stakeholder engagement [13,15] empirical studies specifically examining intersectoral collaboration in Ghana's zoonotic disease surveillance and response (ZDSR) system are still emerging. Although some studies explore high-level One Health policy coordination in Ghana [16], little is known about how intersectoral collaboration occurs at the operational level within ZDSR systems. Given Ghana's high zoonotic disease burden and under-resourced health systems [17–20], effective intersectoral collaboration is essential to make the most efficient use of limited resources and prevent duplication of efforts, making it critical to understand what facilitates or hinders collaboration in practice [21,22].

This study examines the factors that drive or hinder intersectoral collaboration in ZDSR at the operational level in the Greater Accra Metropolitan Area, Ghana. It explores stakeholder perspectives on collaborative experiences, motivations and expectations to identify key influences that can strengthen intersectoral collaboration. By examining the realities of operational-level intersectoral collaboration, this study aims to provide evidence-based insights for developing training programs, policy adjustments, and collaboration strategies that are both effective and sustainable.

Without this empirical foundation, policy initiatives alone may not translate into meaningful improvements in practice [21,22].

## Methods

### Study setting

This study was conducted as part of a larger research project assessing Ghana's current ZDSR systems across three metropolitan areas (Greater Accra, Kumasi and Tamale) and their potential convergence points for optimizing intersectoral collaboration at the operational level. The broader project comprised multiple objectives including: (1) a rabies sectoral surveillance evaluation conducted in select districts in each area; (2) in-depth examination of current intersectoral collaboration mechanisms [23] and (3) identification of barriers and opportunities for operational-level One Health implementation. This paper uses the interview dataset from objective 2 of the study to identify specific factors influencing collaboration dynamics among human, domestic animal and wildlife health sectors in the Greater Accra Metropolitan Area (GAMA).

Given the complex nature of intersectoral collaboration, qualitative research methods offer deeper insights into stakeholder perspectives, institutional culture, and governance structures, uncovering aspects that quantitative approaches may overlook [24–27]. For this study, stakeholders refer to human, animal and wildlife health practitioners who play a direct role in detecting, reporting and responding to zoonotic threats. The term 'stakeholders' may be used interchangeably with 'actors' 'participants' or 'professionals' throughout this study.

Key institutions included the Ghana Health Service (GHS), the Veterinary Services Department (VSD) and the Wildlife Division (WD) of the Forestry Commission. Participants were selected from the district and subdistrict level of GHS and VSD. However, unlike GHS and VSD, the Wildlife Division did not have district-level officers, thus participants were drawn from the regional and national level. While the GHS and VSD officers are based at the district level, their surveillance and response activities span from community to district levels, making them key informants for understanding intersectoral dynamics in practice. Although site selection was determined by the broader project, the inclusion of over half of GAMA's districts provides sufficient scope to examine intersectoral collaboration dynamics and inform the study's conclusions.

### Study design and participant selection

This qualitative study employed semi-structured in-depth interviews to explore factors influencing intersectoral collaboration in ZDSR. Although the GAMA area comprises 25 districts, only 21 had both human and animal health officers necessary for examining intersectoral collaboration across sectors.

A combined purposive and convenience sampling approach was used. Districts and district heads were purposively selected based on their authority and direct involvement in ZDSR, while convenience sampling allowed authorities to either participate directly, nominate subordinates, or involve additional team members depending on availability, provided all participants had direct ZDSR responsibility. Participants were recruited for this study from 21/12/2022 to 31/05/2023.

Following ethical approval, initial invitations were sent to all 21 eligible districts. Due to incomplete or outdated contact information, invitations successfully reached 20 human health district heads and 17 animal health district heads (one of whom oversaw two districts). Non-respondents received, a follow-up call to introduce the study, explain its purpose, seek their participation and refer them back to the email for full details.

In total, 17 human health and 17 animal health district heads confirmed participation, representing 19 districts. Ultimately, interviews were conducted in 16 districts: human health participants were interviewed in 13 districts, animal health participants in 10 districts, with only seven districts having both sectors represented. The reduction from confirmed to interviewed districts occurred due to data saturation, scheduling conflicts, participant availability and time constraints. All three wildlife health participants contacted agreed to participate and were interviewed.

A total of 66 participants were interviewed which corresponds to 41 interviews. Most participants were interviewed alone (26 interviews) but participants from the same sector and district could opt for group interviews which comprised a maximum of 4 people (15 interviews). A total of 37 interviews were conducted in person, two via zoom and two over the phone. The human health sector contributed 43 participants, including medical directors, physicians, public health officers, disease control officers, and public health nurses. The animal health sector provided 20 participants: veterinarians, para-veterinarians, one recently retired animal health practitioner (included due to the considerable institutional knowledge of this person), two laboratory veterinarians and three agriculture officers (who were invited by the veterinarians during interviews). Although laboratory veterinarians operate at the regional and/or national level, they were included due to their essential role in supporting operational-level ZDSR activities through diagnostic services and their direct collaborative relationships with operational-level practitioners. The wildlife health sector included three participants: one national level and two regional level wildlife veterinarians from the Wildlife Division of the Ghana Forestry Commission. Most participants (86%) had over five years of experience in ZDSR which ensured substantial exposure to intersectoral collaboration.

District-level participants were distributed across all 16 districts, with the majority of districts contributing between 1–4 participants per sector, ensuring balanced representation. Animal health teams were small at the district level (typically 1–4 officers per district), and in most cases, all available animal health officers were interviewed, providing near-complete sectoral representation. For human health sectors, which have larger teams, key personnel with direct ZDSR roles were purposively selected. One district's human health sector, however, contributed a larger number of participants (n = 16) due to its involvement in the rabies surveillance evaluation of the broader research project. This evaluation required visiting all public health facilities in that district, which enabled broader sampling of ZDSR personnel beyond the typical 1–4 key stakeholders interviewed in other districts.

## Data collection – Key informant interviews

The interview guide (S1 Appendix) incorporated two frameworks: (1) The Evaluation of Collaboration in a multisectoral surveillance system (ECoSur) tool which details organizational (at both governance and operational level) and functional attributes essential for effective collaboration in multisectoral systems [28]; and (2) The Components of Surveillance and Response Systems for Monitoring and Evaluation Framework of the WHO which outlines key indicators for assessing communicable disease surveillance and response systems [29].

The interview guide was designed to capture participants' experiences with intersectoral collaboration in ZDSR, including their motivations and expectations for collaboration, as well as the barriers and facilitators influencing their ability to collaborate. Probing questions explored reasons for collaboration or its absence. While the study primarily focused on the human, animal and wildlife sectors, participants were encouraged to share collaborative experiences involving any other sector, provided it was within the context of ZDSR.

All interviews were scheduled over the phone but only after the initial email. Before each interview, written consent was obtained, and confidentiality was assured. All interviews were conducted in English and audio recorded with participants' explicit consent. Participants were allowed to pause the recording at any time, such as when discussing sensitive information.

## Data analysis

Interviews were pseudonymized, and then transcribed by professional third-party transcribers. Data were analysed using Reflexive Thematic Analysis (RTA), a qualitative technique for systematically identifying and interpreting patterns within interview data [30]. The analysis followed Braun and Clarke's six step framework:

(1) Familiarisation – we ensured transcription accuracy through multiple readings of the transcripts against the original audio recordings;

(2) Coding – initial coding was done with MAXQDA Analytics Pro 2022 software [31];

(3) Generating themes – we grouped similar codes into themes;

(4) Reviewing themes – we refined and merged overlapping themes;

(5) Defining themes – final themes were identified from the refined codes and organised to ensure coherence and relevance;

(6) Producing the report – themes were synthesized into the study's findings [30].

Inductive coding was used to allow themes to emerge from the data. This included both explicit responses to reasons for collaboration (or lack thereof) and more nuanced, implicit insights. A second independent re-coding was performed to identify any new or missing codes.

All themes reported in this manuscript were derived from multiple participants' discourse; no single theme was based on one individual's perspective. This approach ensures that the analysis reflects shared patterns across the sample rather than isolated comments.

### Ethical considerations

Ethical approval for the main project which included all sub-objectives was obtained from the University of Bonn – Centre for Development Research Ethics Board (22c_Joannishka Dsani), and the Ghana Health Service Ethics Review Committee (GHS-ERC: 023/09/22). Formal permissions were granted by the Veterinary Services Directorate of the Ministry of Food and Agriculture and the Wildlife Department of the Forestry Commission of Ghana.

### Participant confidentiality

To maintain transparency while protecting participant anonymity, quotes are labelled only by sector (HH = human health, AH = animal health, WI = wildlife health). Many participants shared contextually specific experiences that could reveal their identity to readers familiar with the local health system. Once identified through one distinctive quote, any identifier system would allow readers to attribute all other comments made by that same participant. This risk is particularly acute for animal health participants, as most districts have only one professional in this sector. Unique identifiers, district codes, or other demographic descriptors were therefore not used. Direct quotes from wildlife health participants were not included given their very small number (n = 3). Their perspectives are reflected in the analysis and echoed through shared themes with other sectors, but not illustrated with individual quotations. Finally, to avoid overrepresentation, human health quotes from the district with the highest participation (n = 16) were excluded from illustrative examples presented in this manuscript (though included in the analysis), while animal health quotes from this district were retained. Overall, the illustrative quotes presented in the manuscript represent participants from 15 of the 16 districts and across all organisational levels. One district is not represented in the quotes because only one sector was interviewed in that district and their verbatims were not optimal for illustrating the themes identified.

### Results

We found that intersectoral collaboration in ZDSR in GAMA is influenced by two categories of factors: individual factors and structural factors. Individual factors are driven by personal characteristics, relationships, and individual actions that shape collaboration decisions. Structural factors represent systemic conditions within the ZDSR framework that influence collaboration independent of any single individual's efforts. A third category, Positive Collaborative Outcomes, is addressed separately as these factors are consequential, arising from successful collaboration and operating across both the individual and structural domains to create reinforcing cycles. Across these categories, some factors acted primarily

as facilitators, others as barriers, and some could function as either depending on the degree to which they were present or absent. Additionally, participants shared aspirational expectations about what they hoped future collaboration could achieve, representing hoped-for benefits rather than demonstrated influences on behavior. The findings reflect participants' perceptions and lived experiences rather than objectively measured causal relationships. Quotes have been lightly paraphrased for clarity and readability with full non paraphrased illustrative quotes provided in S1 Table.

### Individual factors

**Interpersonal relationships.** Personal rapport between actors was one of the strongest enablers of collaboration. Friendly relationships fostered trust, encouraged communication, and facilitated informal information exchange. These informal collaborations were sometimes so central that their cessation could halt routine data sharing, highlighting their fragility. Shared experiences, such as being new to a district, also created a sense of camaraderie that translated into joint work. One human health participant underscored the importance of personal connection: "*the truth is Dr. [AH] and I are very good friends. There's a personal touch. He and I are the newest in the assembly, […] we work hand in hand, [in] everything*" (HH).

Beyond pure friendship, some relationships were built on reciprocal arrangements where professional services created ongoing, strategic partnerships. This was especially relevant for interacting with high level administrative personnel, such as the District Assembly Coordinator (DAC). In Ghana, the DAC is the administrative head of the District Assembly, responsible for coordinating its activities, implementing government policies and managing resources. The DAC is a key administrative gatekeeper. One animal health participant described using veterinary expertise as a tool for relationship building with the DAC: "*The coordinator, had a [dog] […]so I treated his animals for him and because he liked animals, whenever it came to vaccination he was there to help. The trick we used was that, most of the coordinators had dogs and animals, so we made sure that we treated their animals for them. So, whenever we put in our application, it becomes easier*" (AH). These arrangements blurred the line between relationship-building and transactional collaboration, creating networks that served both personal and professional purposes.

**Personal initiative and assertiveness.** Collaboration was frequently facilitated by individual initiative and proactive engagement. Stakeholders who reached out, asked questions, or introduced themselves were seen as central to starting and sustaining partnerships fostering a sense of reciprocal effort and shared responsibility. One participant, who had received a call from an environmental health colleague during the interview, illustrated this mutual engagement: "*Right now I have 'environment' here [on the phone] …she needs some information, she'll ask me. I need something, I will ask her. That effort is made from both sides. It isn't the same with animal [health]*}" (HH). Deliberately building these relationships often required assertiveness as one animal health participant described: "*I'm vocal. I went there during COVID [and] introduced myself. I was introduced to the whole team. So that rapport was set up. I'm inquisitive and that is how scientists should be. You should investigate, you shouldn't coil back and think they will call you; they won't call.*" (AH).

Conversely, some participants expected others to make the first contact, creating standoffs that prevented collaboration. As one human health participant stated: "*What are the animal health people too doing to collaborate with us? They should also contact us*" (HH). This reluctance to initiate contact, combined with expectations that the "other side" should take responsibility, created barriers to establishing new collaborative relationships.

**Individual motivations.** Participants' motivations strongly shaped whether and how they engaged across sectors. Many described collaborating for practical benefits, such as easier access to data or a wider geographic reach, which improved the efficiency of their work. One animal health actor explained the utility of wider networks: "*If you work in tandem with them [district agriculture officers], you will get benefits, it will help you because it's not everywhere you can reach but they are many*" (AH). In addition to these practical benefits, professional recognition was also a motivator, as some participants noting that collaborative activities enhanced the professional standing or image of the officer, their unit or institution.

Compliance motivations were also evident. Some participants were guided by institutional norms and training to involve relevant stakeholders, while others engaged to meet specific documentation requirements and avoid bureaucratic problems. For instance, animal health officers were legally required to obtain signatures from the National Disaster Management Organization (NADMO) for avian influenza carcass destruction documentation: "*Because on your destruction form, you need NADMO to sign. I did one without informing him [NADMO officer]. Ask him how we suffered*" (AH).

However, not all motivations were framed positively. Some participants admitted to collaborating primarily for self-protection: "*We will call [AH] just to cover my back*" (HH). Others described collaboration as reactive and crisis-driven, with engagement occurring only after problems arose: "*When they go and meet a stumbling block then they call you [asking], 'so what should we do next'?*" (AH).

**Professional hierarchy and status perceptions.**  Perceived professional hierarchies were reported as barriers to collaboration. Several animal health participants felt human health counterparts viewed themselves as superior. Statements like "*the medics feel they're tin-gods*" (AH) and "*[they] always rate themselves higher*" (AH) illustrate this perception. Some traced this attitude and the perception of their own profession's lower status back to the influence of professional training, as one participant explained: "*Right from the university, they will tell you that we are disappointed medical doctors*" (AH). In a similar vein, other participants expressed feeling side-lined, stating: "*… the health, the environmental are really working together, but they side-line us, they don't see the importance of the vets*" (AH). These attitudes discouraged engagement and reinforced sectoral divides.

**Shared interests and alignment.**  Common interests and aligned priorities facilitated collaboration, while their absence hindered it. A critical enabler was a shared interest in public health issues at the leadership level. One human health participant described how the Municipal Chief Executive's (MCE) genuine interest in public health facilitated efforts. The MCE is the executive head and highest political representative of the Municipal/District Assembly, giving them substantial authority over local resources and policy priorities. The participant noted: "*The MCE is very much interested in public health. …anything health, our recent annual review - he stopped everything he was doing and came to sit through and he was taking notes and making remarks and making suggestions*" (HH).

In contrast, misaligned priorities created barriers. Some participants saw little value in collaborating with certain sectors, while others described situations where colleagues verbally agreed to joint plans but subsequently failed to follow through. "*I made sure they [environment health] were part of [the District Health Committee]. However, the issues raised there, nothing. They won't do it; it doesn't concern them. They will say yes at meetings to decisions but they won't do it.*" (HH). This passive resistance, driven by the perception that the task did not concern them, undermined collaborative efforts.

## Structural factors

**Financial resources.**  Financial constraints emerged as a pervasive barrier to intersectoral collaboration across all sectors. Expectations of remuneration for surveillance activities created particular challenges, with participants describing how financial demands discouraged joint efforts. One animal health participant explained how collaboration ceased when payment expectations arose: "*I used to call them [HH] to accompany [me] but nowadays when you go with them, you must give them something. I told them, ' I'm a small boy, I don't have money to be giving to them.'. I use my own money for fuel, and nobody gives me anything*" (AH). These financial expectations occasionally created tension that participants believed hindered future joint engagements.

Resource disparities between sectors further complicated collaboration. The animal health sector's lack of basic infrastructure made meaningful partnership difficult, as illustrated by one human health participant: "*This is the problem. They don't have a place to even keep a dog for observation. We let them know [about dog bite cases] but they are not going to do anything about [it]*" (HH). Shared scarcity of operational resources meant that financially stronger sectors were unable to absorb the costs of weaker ones, halting joint efforts. Participants described situations where collaboration depended

on logistics, such as fuel or transportation, that the requesting sector could not provide, creating a barrier even when both parties were willing to collaborate.

A critical, intertwined barrier was the perception that certain sectors were primarily motivated by financial gain, which fostered mistrust and discouraged initiative. One human health participant noted that environmental health actors '*only come in when there is money*' and were otherwise '*not bothered*' if activities did not '*directly benefit them*' (HH). This mistrust translated directly into reluctance to initiate joint projects. The same participant explained their refusal to write a collaborative funding request, saying they would not do so because environmental health would *"go and take the money and you'll not get the money to do the work."* (HH).

**Workforce availability.** Severe understaffing in animal and wildlife health sectors significantly limited collaboration opportunities. Wildlife officers were entirely absent at the district level, while some districts lacked veterinary officers completely. One participant emphasized this disparity: *"How many veterinary officers are there? Exactly…[…] it's just one person - how much can they do? But as for environmental officers, they are [plenty]."* (HH).

Even where veterinary officers existed, they often managed multiple districts, stretching resources thin and reducing availability for joint activities. Furthermore, restructuring of GAMA's administrative districts led to an uneven personnel distribution, leaving some areas without veterinary officers: *"maybe because the district got separated, and we didn't have a full complement of the veterinary staff"* (HH).

**The governance and organization of surveillance and response.** Surveillance and response system organization created multiple barriers to collaboration. The absence of formalized intersectoral structures meant engagement remained sector-specific rather than integrated. As one participant explained: *"See, in reality, the systems must be in place because we have systems for reporting. But if my system of reporting, doesn't include somebody there's no way I'll go and root that person [out]"* (HH). As a result, collaboration often occurred only on demand, i.e., when triggered by specific events rather than as part of routine surveillance.

Participants expressed frustration over a lack of political will from district assemblies, which hindered collaboration efforts. Even when actors initiated discussions and submitted plans, authorities failed to act:

"This rabies thing for instance, Mr. [AH] and myself met them, we spoke about culling some of the dogs because we have a huge stray dog problem in this district. We talked and talked, they asked him to come up with a plan, he did it, nothing happened. The assembly didn't make any effort." (HH)

This lack of responsiveness discouraged continued advocacy efforts by the personnel involved.

Institutional continuity also suffered, with collaborative structures dissolving when personnel changed: *"this committee [Public Health Emergency Management Committee] was formed just last year, it should have been in existence long ago, but when the people retired or were transferred out, it was never reconstituted"* (HH).

**Familiarity and institutional visibility.** Collaboration was more likely to occur when actors knew of each other's existence, mandates and contact information. Initial introductions and personal interactions were a precursor to establishing routine communication and ongoing collaboration. Proactive engagement in meetings helped establish contacts, which facilitated the transition to information sharing and inclusion in critical structures. One animal health participant described the foundational step of establishing contacts: *"When I came here, I was sent to everybody, we introduced ourselves. So now anything that they are doing, she [HH] would just give me a heads up"* (AH). Conversely, collaboration was sometimes initiated only after an accidental discovery within an unrelated institutional setting revealed a discrepancy of need for specific expertise. This highlights how formal intersectoral spaces could serve as vital discovery mechanisms when routine communication fails: A human health participant described how the Assembly's budget discussion led to a vital first contact with the veterinarian:

"We were discussing budgets then I heard that they had used some money to vaccinate dogs. I asked the number of dogs they vaccinated and juxtaposed that […] So, it's not like I even routinely knew; it was just because we were discussing budgets[…] I came and looked at the dog bites [data] later and then I called him that, 'let's talk'. [and] invited him here - the Vet" (HH)

The visibility of a sector's workforce and its physical presence emerged as a major factor influencing collaboration. Sectors with a large and well-distributed workforce enjoyed greater visibility, making collaboration easier: "*You see why the environmental officers come in all the time? Because they are the people we get to all the time. They are always everywhere*" (HH). Conversely, limited sectoral visibility was a barrier to collaboration. Participants simply did not engage with actors they were unfamiliar with, did not know how to contact, or were unaware of their existence. This lack of visibility was particularly pronounced for animal health actors, with multiple human health participants stating they had never encountered the veterinary officer in their district. A human health participant questioned the fundamental lack of accessibility and contact information for the animal and wildlife health sectors: "*so they too where are they? Who is the focal person? Environment, we know who to reach. Why is it that [for] the wildlife, we don't know who to reach?*" (HH).

This visibility barrier was exacerbated by the animal health sector's limited or absent physical and the lack of basic infrastructure, which reduced recognition and accessibility. Animal health participants emphasized the necessity of a physical office to normalize their presence and increase recognition: "*If we have an office, it's an office you can come to. …, it will make it much easier and our presence will be known.*" (AH). However, the study also noted instances where different sectors shared the same physical premises or office block but had no history of collaborative activities. Participants in these settings described a lack of formal roles or mandates that would require them to interact across sectoral lines.

**Knowledge and training systems.** Public health and One Health knowledge was believed to enhance and encourage intersectoral collaboration. Formal training programs enhanced collaborative capacity, with participants noting how trained individuals facilitated intersectoral engagement: "*So those who were trained, who knew about rabies because of the GFELT (Ghana Field Epidemiology and Laboratory Training) program, or their knowledge in public health, were able to call their colleagues in vet*" (AH). Public health training also appeared to influence collaborative attitudes, with trained professionals described as showing greater respect for other sectors:

"I've noticed that, the medics, those who have done public health, their attitude is different. Those I was having problems with [are] those who have not done public health. After doing public health for two years, when they come to me, they respect me" (AH)

Furthermore, participants believed that proactive knowledge sharing improved willingness to collaborate and drove essential changes in behaviour: "*I remember when I went to the hospital first, I started educating the doctors, and […] then they came to realize that, this thing that we are doing, we should always involve the vets*" (AH).

However, systemic knowledge gaps created barriers. Several participants acquired One Health concepts through personal initiatives rather than institutional training, demonstrating a failure in formal training systems: "*I learnt of this One Health thing through my [own] learning … I heard it once, … But to say that we've had training on it, no*" (HH).

Sectoral silos persisted due to limited training and education about other sector's roles and responsibilities in ZDSR. This lack of educational exposure to the work and relevance of other sectors meant that collaboration with certain sectors was simply unimaginable to some participants:

One participant admitted they had never considered collaboration with the wildlife sector until the interview:

"Wildlife? They are even out of the picture …People have pet monkeys but to think of working with wildlife, I'm telling you the truth, it hasn't even occurred [to me] …. I do [collaborate] with all, except wildlife but you've given me a new perspective" (HH)

                                                                 

**Positive collaborative outcomes.** Unlike the previous factors that fell primarily within individual or structural categories, positive collaborative outcomes from past partnerships operated across both. Participants described concrete benefits that not only encouraged continued collaboration but also created reinforcing cycles, where past success made future engagement more likely. In this way, outcomes bridged individual efforts and structural conditions, becoming predictive factors for sustained collaboration.

Improved response and control measures emerged as an important outcome of collaboration. Participants described how information shared across sectors triggered timely interventions. One human health participant explained how reporting a dog bite case led to district-wide vaccination efforts: "*When we inform, I remember [a recent dog bite case]... they were able to purchase vaccines and then immunized some of the dogs*" (HH). The phrase *"when we inform"* suggests that this process has occurred repeatedly, creating a reliable expectation that reporting will lead to action. This dependability was incorporated into participants' decisions to collaborate.

Enhanced surveillance and early detection were other outcomes participants linked to collaboration. Relationships across sectors enabled alerts that might otherwise have been missed. One animal health participant noted: "*There have been a lot of cases where, because of [our] collaboration, we've been prompted. The cat case that we had; it was the health people that called me... I would not have even heard of the case*" (AH). The reference to *"a lot of cases"* indicates that collaboration provided a consistent and dependable mechanism for identifying threats early. This reliability created mutual dependencies that encouraged participants to maintain these connections.

Improved access to resources, particularly post-exposure rabies vaccines, was also identified as a tangible benefit of intersectoral collaboration. Human health participants described the animal health sector's ability to provide vaccines as especially valuable: "*Veterinary has been helpful as far as dog bite and rabies is concerned. Once we send someone there, they give the vaccine [for] free*" (HH). The confidence in *"once we send someone"* shows that this support has become a dependable arrangement rather than a one-off occurrence. The predictability of this resource sharing provided a practical incentive to sustain collaboration. These experiences demonstrate how positive outcomes become influential factors; participants who experienced benefits were more likely to engage in future collaboration, creating self-reinforcing cycles of engagement.

All factors discussed above are illustrated in Fig 1, which provides a visual summary of the individual and structural factors that influence intersectoral collaboration in ZDSR.

*The Venn diagram illustrates individual contextual factors (orange circle) and structural contextual factors (blue circle) that influence intersectoral collaboration. Positive outcomes from successful collaborations (green, positioned at the intersection) operate across both dimensions, creating reinforcing cycles that sustain engagement. Green ovals represent factors that exclusively facilitate collaboration, red ovals represent barriers, and yellow ovals represent factors that can function as either facilitators or barriers depending on their presence, absence, or degree. Individual factors are driven by personal characteristics and actions, while structural factors represent systemic conditions that operate independently of individual efforts.*

## Stakeholder aspirations for future collaboration

While the factors discussed previously reflect the conditions and experiences shaping current collaboration, participants also expressed aspirational expectations about what intersectoral collaboration could achieve in the future. These perspectives represent hoped-for benefits rather than demonstrated influences on collaborative behavior.

Participants consistently anticipated that collaboration would improve efficiency and effectiveness by leveraging different sectoral strengths, leading to smoother coordination and enhanced disease surveillance. They expected collaboration to enable early detection and prompt action, with the ultimate goal being holistic population protection: *"The ultimate goal is to protect the public, animals included. Because if you control it from the animal's side, that is when the public [can benefit]"* (AH). The COVID-19 pandemic reinforced participants' conviction that intersectoral collaboration is essential for

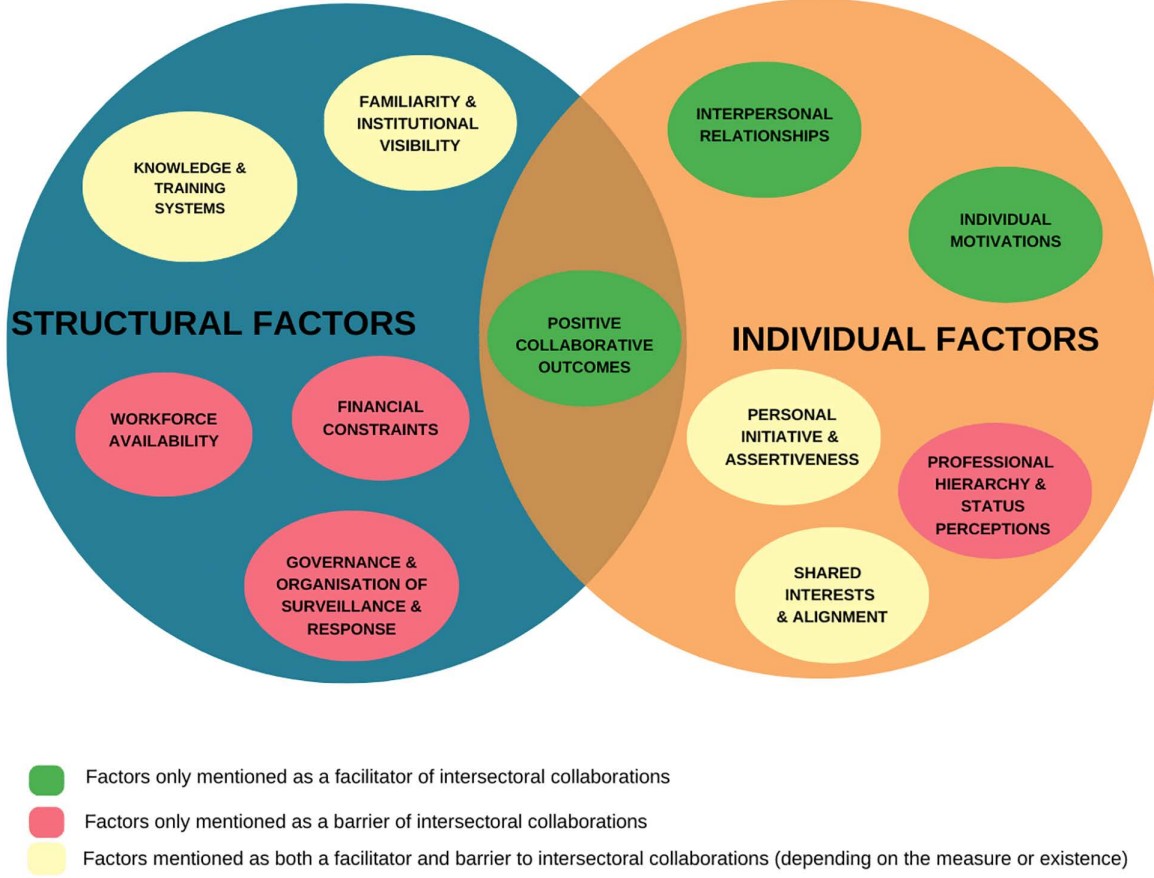

Legend:

- Factors only mentioned as a facilitator of intersectoral collaborations
- Factors only mentioned as a barrier of intersectoral collaborations
- Factors mentioned as both a facilitator and barrier to intersectoral collaborations (depending on the measure or existence)

**Fig 1.  Conceptual framework of factors influencing intersectoral collaboration in zoonotic disease surveillance and response in Greater Accra Metropolitan Area, Ghana.**

ZDSR. One participant expressed hope that the pandemic would catalyze greater recognition of the animal health sector's critical role: *"And sorry to say, but I'm happy that the pandemic is happening and it is all originating from animals. So now, the world will sit up and take this sector serious"* (AH).

A key aspiration was leveraging intersectoral collaboration as a platform for public awareness and education. Participants believed that a coordinated, unified voice was necessary to raise the visibility of zoonotic disease risks and prevention among the general public. Beyond public education benefits, participants also viewed collaboration as a platform for learning, skill-sharing, and capacity building for themselves (the professionals), anticipating mutual learning benefits that improve future independent performance:

"The more outbreaks or surveillance work you are doing, the more you are improving your knowledge. […] you collaborate and at the end of the day, you have ideas on how to handle another[case]. Even if the [other sectors] were not there, you would handle it better [because of] prior experience." (HH).

Resource sharing aspirations focused on overcoming sectoral limitations through systematic pooling, envisioning coordinated multi-sector responses based on past successful experiences. One participant provided a practical example of

this vision in action during a crisis: "*Collaboration is good because I remember… we had a bird flu [case]. We went to the Fire Service people for their fire tender to do the depopulation*" (AH).

Finally, participants aspired to achieve holistic, intersectoral approaches to disease control that address environmental and policy dimensions. They envisioned comprehensive surveillance systems where various sectors coordinate to track disease risks. One participant described an elaborate vision of collaborative disease prevention driven by integrated surveillance and alerting:

"If we collaborate well, we'll be able to reduce episodes of outbreaks. [...]Imagine if the people in the wildlife start tracing. [...] Then they can alert the assembly […] Then their environment (unit) starts looking […] [Then human] health can also start community engagement.. That way, we will prevent anyone from getting the disease" (HH)

Participants also envisioned collaboration driving policy enforcement for disease prevention. They anticipated that intersectoral engagement would enable local government bodies to enforce existing bylaws, as one participant illustrated with a hypothetical scenario: "*Animal [health] can tell assembly that 'this group of animals [e.g., stray dogs] are vectors... we shouldn't allow them to be at residential areas.' Then assembly [will] ensure that their bylaws are enforced, because now, they [understand] the reasons behind it*" (HH)

These aspirational visions reflected participants' understanding of ideal intersectoral collaboration, even when current structural constraints prevented their realization.

## Discussion

This study reveals that intersectoral collaboration in ZDSR is shaped by the interplay of structural deficits, individual agency, and feedback from collaborative experiences. The central challenge is the absence of formal institutional frameworks, which forces collaboration to rely on informal, fragile mechanisms. This manifests in three interconnected ways: inequitable access determined by personal networks rather than expertise; perceived professional hierarchies reinforced by unclear roles; and pragmatic rather than idealistic motivations for engagement. However, successful collaborative experiences create reinforcing cycles that sustain participation, demonstrating collaboration's potential value and providing compelling rationale for formalization.

### The interplay between individual agency and systemic structural barriers

In the absence of formal structures, personal networks become the primary drivers of collaboration. While interpersonal rapport facilitates rapid coordination where formal systems are weak [32,33], reliance on informal arrangements creates three interconnected barriers:

First, inequitable access to collaborative opportunities. Participation depends on who knows whom rather than relevant expertise. Systems requiring constant self-advocacy privilege assertive individuals while systematically excluding others [34,35], burdening marginalized sectors to repeatedly prove their worth and reinforcing the inequities One Health aims to address.

Second, fragility. Informal networks lack standardized procedures or institutional memory, making collaboration vulnerable to staff turnover. Even where committee reconstitution requirements exist, weak enforcement allows structures to dissolve during personnel transitions.

Third, perceived hierarchies. Participants reported perceived human health dominance likely stemming from visible disparities in workforce capacity, and resources, given that the human health sector is typically better resourced [36–38]. Without clearly defined cross-sectoral roles, actors' default to interpreting hierarchies subjectively, creating barriers even where objective power differences are minimal.

Beyond relational dynamics, structural fragmentation creates operational barriers. Financial mistrust, with participants fearing resource exploitation, discourages collaborative initiatives [39,40]. Weak governance and limited support from

district authorities fragment efforts. Event-driven rather than routine collaboration lack of follow-through on submitted plans (such as rabies control proposals) disrupt continuity and limit the institutionalization of collaboration [21,25,41].

Workforce scarcity compounds these problems: limited veterinary and wildlife health professionals create capacity issues and reduce sectoral visibility [42–44]. Human health actors were often unaware animal health counterparts existed, while others had never even considered collaborating with wildlife health. Physical inaccessibility further reduced visibility. While research suggests that physical proximity increases collaboration likelihood [45], this study observed instances where sectors shared the same office building but had never collaborated due to absent formal mandates and defined roles. This demonstrates that adequate staffing and physical presence are essential for sectoral visibility and accessibility, and while proximity may create opportunities for collaboration, these conditions remain insufficient without formal institutional frameworks to mandate and structure engagement.

## Pragmatic drivers and virtuous cycles of engagement

This study challenges assumptions that One Health collaboration is driven by shared, collective goals. Engagement is often motivated by pragmatic concerns: instrumental benefits and 'covering one's back' to avoid professional setbacks. When participation is determined by institutional incentives rather than proactive choice, collaboration remains opportunistic and unsustainable [12,46–49].

However, successful experiences create reinforcing cycles. Participants experiencing concrete benefits, improved outbreak response, early disease detection, enhanced resource access, were more willing to engage in future collaboration, a dynamic underexplored in One Health literature which emphasizes barriers [21,25,41]. When collaboration delivers visible value, expectations of future benefits increase.

Yet these cycles require consistent institutional support. Failed collaborative attempts, such as when sectors lacked facilities to follow through on commitments, discouraged future engagement. This creates an expectation-reality gap: participants held aspirational visions of improved efficiency and comprehensive resource-sharing that remained largely unrealized, contributing to frustration and disengagement [21,50–52]. Initial positive experiences cannot substitute for structural conditions enabling consistent benefit delivery [16,53].

## Structural solutions – Optimizing institutional architecture for one health

Addressing these challenges requires modifying existing institutional structures rather than creating new One Health platforms, which often remain non-functional [54,55]. While complete resource equity is unlikely in resource-constrained contexts where human health receives greater political attention, modest structural adjustments can mitigate power imbalances.

The district Public Health Emergency Management Committee (PHEMC) represents an ideal existing platform for this approach. Established under GHS's Integrated Disease Surveillance and Response (IDSR) guidelines [56], PHEMCs are intended as multisectoral coordinating bodies for surveillance and emergency preparedness, with mandated regular meetings during non-emergency periods. However, implementation remains inconsistent. Participants described committees dissolving after staff transitions, and many human health actors had never met their veterinary counterparts despite working in the same district for years. These gaps reflect structural ambiguity in PHEMC membership guidelines. The IDSR specifies the "District Director of Health Services" and "District Environmental Health Officer" as distinct members, yet veterinary services are designated as "District Director of Veterinary/Agricultural Services" [56], grouping animal health expertise with agricultural leadership. This conflation creates uncertainty over representation and provides no mechanism to ensure veterinary participation during committee reconstitution. Clarifying PHEMC membership to explicitly mandate "District Veterinary Officer" and enforcing regular meetings would address multiple barriers simultaneously: routine meetings would institutionalize contact (mitigating unfamiliarity); formal inclusion would remove reliance on personal relationships or self-advocacy; and planning platforms would reinforce collective ownership.

Simple administrative measures could further strengthen collaboration. Participants' experiences varied widely depending on whether new personnel were formally introduced to other sector leads upon posting. District Assemblies should implement a standardized introduction protocol, such as circulating new personnel profiles and contact details via email to all relevant departments, thus ensuring early familiarity and reducing reliance on chance encounters.

Beyond administrative reform, institutionalizing One Health education can neutralize hierarchies. Participants noted that human health professionals with postgraduate public health training demonstrated more collaborative attitudes and greater respect for cross-sectoral expertise than those without. While public health training is officially required for district health leadership [57], some leaders reported self-educating on One Health principles, suggesting that these principles are not yet standardized. Consistently emphasizing One Health principles within preservice curricula and in-service training across all relevant health cadres could normalize collaborative attitudes as a professional standard rather than an individual disposition.

These educational efforts must be complemented by formalized communication and data sharing protocols. Rather than implementing complex technologies, routine exchange of aggregated district-level surveillance reports would maintain shared situational awareness while preserving operational control. Clear distinctions between routine updates and urgent notification triggers for notifiable diseases would proceduralize joint actions. Defining roles and response protocols for specific disease scenarios would further institutionalize this collaboration.

These suggested structural reforms require adequate human resources to function effectively. Environmental health's "visibility" stemmed from workforce numbers, whereas veterinary services were perceived as "just one person" and easily overlooked. Without increased staffing and dedicated infrastructure, such as district veterinary clinics, even well-designed institutional reforms may struggle to achieve an operational impact.

These reforms focus on achievable improvements in coordination and communication rather than comprehensive resource equity, which remains constrained by broader budget realities. By delivering tangible benefits in information exchange, joint planning, and routine engagement, these structural modifications can bridge the expectation-reality gap that currently fuels frustration and disengagement.

Finally, these district-focused recommendations do not address wildlife health marginalization which stems from having only three wildlife veterinarians nationally with no district-level presence. Addressing this gap requires national-level workforce investment and policy intervention beyond the scope of local governance mechanisms.

### Limitations and perspectives

This study has limitations. First, self-reported data may introduce bias and precludes establishing causality or measuring actual impacts. Future research could combine interviews with quantitative assessments of measurable outcomes (e.g., response times, vaccination coverage) to establish clearer links between collaboration and public health results.

Second, while we implemented measures to enhance trustworthiness, including independent re-coding of a subset of transcripts, resource constraints prevented dual coding of the entire dataset, which would have further minimized potential single-coder bias.

Third, this study prioritized in-depth insights into operational-level collaboration dynamics within GAMA over statistical generalizability. While systemic One Health barriers are documented regionally [15], evidence on how these operate within Ghana's specific operational architecture remained limited. This study addresses that gap, offering detailed understanding of dynamics transferable to similar LMIC contexts, though their manifestations and relative importance may vary across settings. Future comparative research across multiple regions and longitudinal studies tracking collaboration over time would strengthen understanding of how these patterns operate in different contexts and evolve with policy interventions.

Finally, while this study focused on domestic governance barriers at the operational level, international funding mechanisms and global health priorities also shape national One Health implementation in ways that may create competition

between national actors or prioritize external agendas over local needs. Understanding how international frameworks interact with domestic collaboration dynamics warrants future research.

## Conclusions

This study reveals that Ghana's One Health collaboration challenge is not primarily insufficient resources or political will, but the absence of institutional frameworks that mandate and structure intersectoral engagement. Without formal mechanisms, collaboration depends on fragile personal networks, creating inequitable access and perceived hierarchies that undermine sustainability. However, positive collaborative experiences generate reinforcing cycles of engagement, demonstrating that when properly supported, collaboration can be self-sustaining.

Three actionable reforms emerge: (1) clarifying Public Health Emergency Management Committee membership to explicitly mandate veterinary officer representation, (2) institutionalizing One Health principles within health professional training, and administrative protocols (3) investing in veterinary and wildlife health workforce capacity. These pragmatic modifications leverage existing governance structures rather than creating new platforms, addressing the operational realities documented in this study.

By examining how district-level collaboration actually functions, rather than how policies envision it should function, this study provides evidence-based insights essential for translating One Health aspirations into functional practice. Our findings offer transferable lessons for LMICs facing similar operational complexities, highlighting that sustainable collaboration requires deliberate structural reform, not reliance on individual champions.

## Supporting information

**S1 Appendix. Interview guide.**
(PDF)

**S1 Table. Supporting quotes for factors influencing intersectoral collaboration in zoonotic disease surveillance and response in the Greater Accra Metropolitan Area of Ghana.**
(PDF)

## Acknowledgments

We would like to acknowledge the support of our colleagues at the One Health Graduate School, Center for Development Research, University of Bonn, Germany. We are also grateful to the authorities and participants from the Ghana Health Service, the Veterinary Services Directorate, and the Wildlife Division of the Ghana Forestry Commission for their invaluable support. Special thanks to Dr. Fenteng Danso (Epidemiology Unit, Veterinary Services Department, Ghana), Dr. Azumah Abdul-Tawab (Public Health Unit, Ghana Health Service, Ghana), and Dr. Meyir Ziekah (Wildlife Division, Forestry Commission, Ghana) for their generous assistance.

## Author contributions

**Conceptualization:** Joannishka K. Dsani, Sherry Ama Mawuko Johnson, Walter Bruchhausen.

**Data curation:** Joannishka K. Dsani.

**Formal analysis:** Joannishka K. Dsani.

**Funding acquisition:** Joannishka K. Dsani, Walter Bruchhausen.

**Investigation:** Joannishka K. Dsani.

**Methodology:** Joannishka K. Dsani, Sherry Ama Mawuko Johnson, Sandul Yasobant.

**Project administration:** Joannishka K. Dsani.

Resources: Sherry Ama Mawuko Johnson, Walter Bruchhausen.

Supervision: Sherry Ama Mawuko Johnson, Sandul Yasobant, Walter Bruchhausen.

Visualization: Joannishka K. Dsani.

Writing – original draft: Joannishka K. Dsani.

Writing – review & editing: Joannishka K. Dsani, Sherry Ama Mawuko Johnson, Sandul Yasobant, Walter Bruchhausen.

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
