## [Decision Letter · Decision Letter 0]

22 Jul 2025

Dear Dr. Dsani,

Thank you for submitting your manuscript to PLOS ONE. After careful consideration, we feel that it has merit but does not fully meet PLOS ONE’s publication criteria as it currently stands. Therefore, we invite you to submit a revised version of the manuscript that addresses the points raised during the review process.

**Reviewer 1**

The study uses a quantitative approach based on a thematic analysis of surveillance actors’ discourse to explore the challenges of operationalizing the One Health concept for the surveillance of zoonotic diseases, through the lens of collaboration across actors, at the sub-national level. Based on the findings of this analysis, the authors claim that they provide insights to strengthen intersectoral collaboration and improve public health outcomes.

The manuscript is well written and easy to read up to the end of the methods section.

Conversely, the format of the results section should be deeply revised. The thematic analysis allows you to identify relevant themes related to your research questions. But, when describing the results, and especially in a scientific article, you need to bring back everything together in a well-articulated text. No need to clearly state Theme 1, sub-theme 1, etc. They are not themes anymore, they are what you are willing to describe, eg factors. You can have one separate part for each key factors and then describe the sub-themes in these parts, mentioning if they are acting as barrier or driver. Then for parts B, C, and D, you should write a descriptive analysis of your factors and not provide the results in a table. You can refer to other articles in that field to see how to better structure your result section. I think this should allow you to group some of the sub-themes together among themes and across themes, among parts and across parts (see my comment in the Results sections about merging the 4 parts as motivation, expectations and benefits can also be considered as factors influencing collaboration). Some seem redundant.

In addition, it is difficult to see how some verbatims are illustrating the text (for instance, verbatim lines 309-313).

There are a lot of redundancies and inconsistencies in the discussion part. I suggest to restructure this part and make your message clearer. There are very interesting and innovative ideas but they are drowned out by repetition of things already said elsewhere in the discussion section or even in the results section. I have provided one example in the comment below, but it applies to other parts of the discussion.

I am also surprised that you address the need to align expectations among stakeholders from the different sectors on one hand, but refer to collaboration outcomes only for public health on the other (lines 29-30). One Health is seeking shared benefits across the human, animal and ecosystem health. Supporting this anthropocentric definition of One Health – as done in this manuscript- is contributing to maintain expectation gaps across sectors.

In general, you need to write more and avoid making lists with bullets or tables with a lot of text that is difficult to read

You use “cross-sectoral”, “cross sector” and “inter-sectoral” to characterize collaboration in your manuscript. If you use them interchangeably, I would suggest to opt for one and keep it all along the manuscript. If they have different meaning, please specify.

I would suggest to keep intersectoral collaboration in full, instead of using an acronym.

Some typos need to be corrected.

Abstract

Line 28 : I would not use district as a synonym of operational – so better to choose between district and operational, and not keep the two side-by-side

Line 35: when you say “governance and surveillance organization”, is organization applied to surveillance and governance or to surveillance only? In both cases, it sounds a bit strange to me. I wonder if “surveillance governance and organization” would not be more suitable.

You announce 9 factors but I count only 8.

Line 39: Please specify the outcomes of what. Collaboration?

Line 45: we usually prefer to use collaboration without s

Introduction

Line 57: I do not see how limited public health infrastructure can favor spillover. Can you clarify?

Line 59: 28 934 is not really an approximation and I am a bit skeptical about our capacity to count cases so precisely, especially in SSA where surveillance systems are weak … I would suggest “approximately 30 000”.

Line 62: I think you could improve you definition of One Health. It is an integrated approach but it does not integrate healths – it recognizes that they are interconnected, which require “integrated” actions.

Lines 63-65: I do not understand this sentence. Please clarify.

Line 67: I would suggest to change “district (operational)” for “sub-national”. And then specify that you are working at the level of the district (and why) in the methodology part.

Line 75 : I would suggest to delete “, or fails” that suggest that you anticipate that it is not working

Line 79. I suggest to add “for zoonotic diseases” after systems.

Methods

Lines 101-102. Please describe briefly these criteria for the reader, so he.she can assess the potential impacts on the study results

Line 119: the number of invitations sent is missing – did everyone accept to be part of the interview? if not, why?

We are also missing the distribution of participants across the 16 districts to assess the level of representativity of each district

Lines 137-138: ECoSur details organizational (at governance and operational level) and functional attributes (and not functional and governance attributes)

Line 138. The “.” should be replaced by a “;”.

Line 154 : I do not get how multiple readings ensure accuracy ; and accuracy of what? please explain

I suggest to put together paragraph lines 80-82 with paragraph lines 91-96, and move the contents of lines 83-90 to the method part.

It is usually advised that the coding is done by two people to improve accuracy. If you did not do it, you need to discuss this potential bias in the limits of the method

Lines 171-178: the contents of these 2 paragraphs should be moved to Study design and Data collection

Results

Why did you address motivation, expectations, and perceived benefits separately from factors driving and hindering collaboration?

To me they are also factors that influence collaboration. And you present them as such at the beginning of the discussion (lines 621-622).

Please clarify and explain how everything is linked, or introduce them as specific themes in part A. and merge everything together

Lines 183: why perceived? they were suggested by participants as factors? It is not what I understood in the methodology. They are perceptions of interviewees that you have identified, with your coding process, as being determinants to collaboration; but to me, they are not perceived factors per se (which suggests that interviewees identified themselves the factors)

Line 186. I do not think that an expectation can be perceived

Line 189: I do not get why assigning a unique identifier to participant will allow to link quotes to specific individuals. And I think the reader needs this information to assess the diversity of individuals you are quoting. It would be even better if you could give a code to the district where the interviewee is working to assess the representativeness of the districts

Lines 196-197: you said that the factors are grouped into 2 categories (structural and individual) but then these categories are not used in the description of the factors. In addition, in the abstract you mention sociocultural, structural and governance factors, but you do not identify these groups in the Result section. You need to harmonize across the text. Then in the discussion part, you mention again the structural and individual factors but we do not know which ones fall into these 2 categories.

Discussion

In the abstract, you mention that levers for better collaborations are stronger governance, equitable partnerships, and realistic alignment of stakeholder expectations. However, in the discussion, we do not identify these 3 main levers.

Lines 633-634. Please explain and not only make references to other articles.

Lines 655-657. You refer to participants discourse but you had references to other studies. This is inconsistent.

Lines 658-659. I do not see the link between the profit-driven dimension of collaboration and resource disparities among stakeholders. Please specify.

Lines 678-683. I do not see how ideas in this paragraph articulates with each other. How will mutual respect and engagement contribute to raise awareness and access?

It is also partly redundant with the contents of the paragraph before.

Lines 684-690 seem redundant with what has been said before (sectoral visibility)

Lines 754. You mention that focusing on Ghana prevents the generalization to other LMICs but in the abstract you state that you results are applicable to other LMICs. Please be clearer on what can be generalized and what cannot.

In addition, you have worked on a limited number of districts in a single region. Can you even generalize your results to Ghana? Please discuss this point.

Conclusions

Lines 771. This is not the first study highlighting the role of informal networks, personal initiative, and structural

Constraints in intersectoral collaboration. This has been described in some of the literature cited in this study;

Line 779. Inconsistent with the limit you have mentioned above, about the generalization of your work (see my comment above).

Line 780. Only public health outcomes? how about animal health outcomes? Economic outcomes? Social outcomes? …

**Reviewer 2**

In this submission to PLOS One, the authors explore the factors influencing collaboration in zoonotic disease surveillance and response at the district (operational) level, providing insights to strengthen cross-sector collaboration and improve public health outcomes. The authors use reflexive thematic analysis to analyze interviews with 66 professionals from the human, animal, and wildlife health sectors, all directly involved in zoonotic disease surveillance and response. The authors find that their study provides insights into these realities, highlighting the need for stronger governance, equitable partnerships, and realistic alignment of stakeholder expectations.

I consider this manuscript to be of interest to machine learning researchers in biological systems as well as readers of this journal. As such, I am generally supportive of publication with a few minor comments. While the authors focus on reflexive thematic analysis, there have been recent studies using advanced machine learning methods, such as t-SNE method, which can be used efficiently for classification as well as detecting trends in biochemical systems, which should be noted: Environ. Sci. Technol. Lett. 2023, 10, 1017–1022 and Mar. Genom. 2020, 51, 100723. In particular, these prior studies showed that t-SNE can give classification of results that could help diagnose trends in biochemical systems. I am not asking the authors to carry out such t-SNE calculations, but these recent applications of t-SNE should also be noted.

We look forward to receiving your revised manuscript.

Kind regards,

Asokan Govindaraj Vaithinathan

Academic Editor

PLOS ONE

Journal Requirements:

Additional Editor Comments:

Reviewer 1

The study uses a quantitative approach based on a thematic analysis of surveillance actors’ discourse to explore the challenges of operationalizing the One Health concept for the surveillance of zoonotic diseases, through the lens of collaboration across actors, at the sub-national level. Based on the findings of this analysis, the authors claim that they provide insights to strengthen intersectoral collaboration and improve public health outcomes.

The manuscript is well written and easy to read up to the end of the methods section.

Conversely, the format of the results section should be deeply revised. The thematic analysis allows you to identify relevant themes related to your research questions. But, when describing the results, and especially in a scientific article, you need to bring back everything together in a well-articulated text. No need to clearly state Theme 1, sub-theme 1, etc. They are not themes anymore, they are what you are willing to describe, eg factors. You can have one separate part for each key factors and then describe the sub-themes in these parts, mentioning if they are acting as barrier or driver. Then for parts B, C, and D, you should write a descriptive analysis of your factors and not provide the results in a table. You can refer to other articles in that field to see how to better structure your result section. I think this should allow you to group some of the sub-themes together among themes and across themes, among parts and across parts (see my comment in the Results sections about merging the 4 parts as motivation, expectations and benefits can also be considered as factors influencing collaboration). Some seem redundant.

In addition, it is difficult to see how some verbatims are illustrating the text (for instance, verbatim lines 309-313).

There are a lot of redundancies and inconsistencies in the discussion part. I suggest to restructure this part and make your message clearer. There are very interesting and innovative ideas but they are drowned out by repetition of things already said elsewhere in the discussion section or even in the results section. I have provided one example in the comment below, but it applies to other parts of the discussion.

I am also surprised that you address the need to align expectations among stakeholders from the different sectors on one hand, but refer to collaboration outcomes only for public health on the other (lines 29-30). One Health is seeking shared benefits across the human, animal and ecosystem health. Supporting this anthropocentric definition of One Health – as done in this manuscript- is contributing to maintain expectation gaps across sectors.

In general, you need to write more and avoid making lists with bullets or tables with a lot of text that is difficult to read

You use “cross-sectoral”, “cross sector” and “inter-sectoral” to characterize collaboration in your manuscript. If you use them interchangeably, I would suggest to opt for one and keep it all along the manuscript. If they have different meaning, please specify.

I would suggest to keep intersectoral collaboration in full, instead of using an acronym.

Some typos need to be corrected.

Abstract

Line 28 : I would not use district as a synonym of operational – so better to choose between district and operational, and not keep the two side-by-side

Line 35: when you say “governance and surveillance organization”, is organization applied to surveillance and governance or to surveillance only? In both cases, it sounds a bit strange to me. I wonder if “surveillance governance and organization” would not be more suitable.

You announce 9 factors but I count only 8.

Line 39: Please specify the outcomes of what. Collaboration?

Line 45: we usually prefer to use collaboration without s

Introduction

Line 57: I do not see how limited public health infrastructure can favor spillover. Can you clarify?

Line 59: 28 934 is not really an approximation and I am a bit skeptical about our capacity to count cases so precisely, especially in SSA where surveillance systems are weak … I would suggest “approximately 30 000”.

Line 62: I think you could improve you definition of One Health. It is an integrated approach but it does not integrate healths – it recognizes that they are interconnected, which require “integrated” actions.

Lines 63-65: I do not understand this sentence. Please clarify.

Line 67: I would suggest to change “district (operational)” for “sub-national”. And then specify that you are working at the level of the district (and why) in the methodology part.

Line 75 : I would suggest to delete “, or fails” that suggest that you anticipate that it is not working

Line 79. I suggest to add “for zoonotic diseases” after systems.

Methods

Lines 101-102. Please describe briefly these criteria for the reader, so he.she can assess the potential impacts on the study results

Line 119: the number of invitations sent is missing – did everyone accept to be part of the interview? if not, why?

We are also missing the distribution of participants across the 16 districts to assess the level of representativity of each district

Lines 137-138: ECoSur details organizational (at governance and operational level) and functional attributes (and not functional and governance attributes)

Line 138. The “.” should be replaced by a “;”.

Line 154 : I do not get how multiple readings ensure accuracy ; and accuracy of what? please explain

I suggest to put together paragraph lines 80-82 with paragraph lines 91-96, and move the contents of lines 83-90 to the method part.

It is usually advised that the coding is done by two people to improve accuracy. If you did not do it, you need to discuss this potential bias in the limits of the method

Lines 171-178: the contents of these 2 paragraphs should be moved to Study design and Data collection

Results

Why did you address motivation, expectations, and perceived benefits separately from factors driving and hindering collaboration?

To me they are also factors that influence collaboration. And you present them as such at the beginning of the discussion (lines 621-622).

Please clarify and explain how everything is linked, or introduce them as specific themes in part A. and merge everything together

Lines 183: why perceived? they were suggested by participants as factors? It is not what I understood in the methodology. They are perceptions of interviewees that you have identified, with your coding process, as being determinants to collaboration; but to me, they are not perceived factors per se (which suggests that interviewees identified themselves the factors)

Line 186. I do not think that an expectation can be perceived

Line 189: I do not get why assigning a unique identifier to participant will allow to link quotes to specific individuals. And I think the reader needs this information to assess the diversity of individuals you are quoting. It would be even better if you could give a code to the district where the interviewee is working to assess the representativeness of the districts

Lines 196-197: you said that the factors are grouped into 2 categories (structural and individual) but then these categories are not used in the description of the factors. In addition, in the abstract you mention sociocultural, structural and governance factors, but you do not identify these groups in the Result section. You need to harmonize across the text. Then in the discussion part, you mention again the structural and individual factors but we do not know which ones fall into these 2 categories.

Discussion

In the abstract, you mention that levers for better collaborations are stronger governance, equitable partnerships, and realistic alignment of stakeholder expectations. However, in the discussion, we do not identify these 3 main levers.

Lines 633-634. Please explain and not only make references to other articles.

Lines 655-657. You refer to participants discourse but you had references to other studies. This is inconsistent.

Lines 658-659. I do not see the link between the profit-driven dimension of collaboration and resource disparities among stakeholders. Please specify.

Lines 678-683. I do not see how ideas in this paragraph articulates with each other. How will mutual respect and engagement contribute to raise awareness and access?

It is also partly redundant with the contents of the paragraph before.

Lines 684-690 seem redundant with what has been said before (sectoral visibility)

Lines 754. You mention that focusing on Ghana prevents the generalization to other LMICs but in the abstract you state that you results are applicable to other LMICs. Please be clearer on what can be generalized and what cannot.

In addition, you have worked on a limited number of districts in a single region. Can you even generalize your results to Ghana? Please discuss this point.

Conclusions

Lines 771. This is not the first study highlighting the role of informal networks, personal initiative, and structural

Constraints in intersectoral collaboration. This has been described in some of the literature cited in this study;

Line 779. Inconsistent with the limit you have mentioned above, about the generalization of your work (see my comment above).

Line 780. Only public health outcomes? how about animal health outcomes? Economic outcomes? Social outcomes? …

Reviewer 2

In this submission to PLOS One, the authors explore the factors influencing collaboration in zoonotic disease surveillance and response at the district (operational) level, providing insights to strengthen cross-sector collaboration and improve public health outcomes. The authors use reflexive thematic analysis to analyze interviews with 66 professionals from the human, animal, and wildlife health sectors, all directly involved in zoonotic disease surveillance and response. The authors find that their study provides insights into these realities, highlighting the need for stronger governance, equitable partnerships, and realistic alignment of stakeholder expectations.

I consider this manuscript to be of interest to machine learning researchers in biological systems as well as readers of this journal. As such, I am generally supportive of publication with a few minor comments. While the authors focus on reflexive thematic analysis, there have been recent studies using advanced machine learning methods, such as t-SNE method, which can be used efficiently for classification as well as detecting trends in biochemical systems, which should be noted: Environ. Sci. Technol. Lett. 2023, 10, 1017–1022 and Mar. Genom. 2020, 51, 100723. In particular, these prior studies showed that t-SNE can give classification of results that could help diagnose trends in biochemical systems. I am not asking the authors to carry out such t-SNE calculations, but these recent applications of t-SNE should also be noted.

Reviewers' comments:

Reviewer's Responses to Questions

**Comments to the Author**

1. Is the manuscript technically sound, and do the data support the conclusions?

Reviewer #1: Yes

Reviewer #2: Yes

2. Has the statistical analysis been performed appropriately and rigorously?

Reviewer #1: N/A

Reviewer #2: Yes

3. Have the authors made all data underlying the findings in their manuscript fully available?

Reviewer #1: No

Reviewer #2: Yes

4. Is the manuscript presented in an intelligible fashion and written in standard English?

Reviewer #1: Yes

Reviewer #2: Yes

Reviewer #1: The study uses a quantitative approach based on a thematic analysis of surveillance actors’ discourse to explore the challenges of operationalizing the One Health concept for the surveillance of zoonotic diseases, through the lens of collaboration across actors, at the sub-national level. Based on the findings of this analysis, the authors claim that they provide insights to strengthen intersectoral collaboration and improve public health outcomes.

The manuscript is well written and easy to read up to the end of the methods section.

Conversely, the format of the results section should be deeply revised. The thematic analysis allows you to identify relevant themes related to your research questions. But, when describing the results, and especially in a scientific article, you need to bring back everything together in a well-articulated text. No need to clearly state Theme 1, sub-theme 1, etc. They are not themes anymore, they are what you are willing to describe, eg factors. You can have one separate part for each key factors and then describe the sub-themes in these parts, mentioning if they are acting as barrier or driver. Then for parts B, C, and D, you should write a descriptive analysis of your factors and not provide the results in a table. You can refer to other articles in that field to see how to better structure your result section. I think this should allow you to group some of the sub-themes together among themes and across themes, among parts and across parts (see my comment in the Results sections about merging the 4 parts as motivation, expectations and benefits can also be considered as factors influencing collaboration). Some seem redundant.

In addition, it is difficult to see how some verbatims are illustrating the text (for instance, verbatim lines 309-313).

There are a lot of redundancies and inconsistencies in the discussion part. I suggest to restructure this part and make your message clearer. There are very interesting and innovative ideas but they are drowned out by repetition of things already said elsewhere in the discussion section or even in the results section. I have provided one example in the comment below, but it applies to other parts of the discussion.

I am also surprised that you address the need to align expectations among stakeholders from the different sectors on one hand, but refer to collaboration outcomes only for public health on the other (lines 29-30). One Health is seeking shared benefits across the human, animal and ecosystem health. Supporting this anthropocentric definition of One Health – as done in this manuscript- is contributing to maintain expectation gaps across sectors.

In general, you need to write more and avoid making lists with bullets or tables with a lot of text that is difficult to read

You use “cross-sectoral”, “cross sector” and “inter-sectoral” to characterize collaboration in your manuscript. If you use them interchangeably, I would suggest to opt for one and keep it all along the manuscript. If they have different meaning, please specify.

I would suggest to keep intersectoral collaboration in full, instead of using an acronym.

Some typos need to be corrected.

Abstract

Line 28 : I would not use district as a synonym of operational – so better to choose between district and operational, and not keep the two side-by-side

Line 35: when you say “governance and surveillance organization”, is organization applied to surveillance and governance or to surveillance only? In both cases, it sounds a bit strange to me. I wonder if “surveillance governance and organization” would not be more suitable.

You announce 9 factors but I count only 8.

Line 39: Please specify the outcomes of what. Collaboration?

Line 45: we usually prefer to use collaboration without s

Introduction

Line 57: I do not see how limited public health infrastructure can favor spillover. Can you clarify?

Line 59: 28 934 is not really an approximation and I am a bit skeptical about our capacity to count cases so precisely, especially in SSA where surveillance systems are weak … I would suggest “approximately 30 000”.

Line 62: I think you could improve you definition of One Health. It is an integrated approach but it does not integrate healths – it recognizes that they are interconnected, which require “integrated” actions.

Lines 63-65: I do not understand this sentence. Please clarify.

Line 67: I would suggest to change “district (operational)” for “sub-national”. And then specify that you are working at the level of the district (and why) in the methodology part.

Line 75 : I would suggest to delete “, or fails” that suggest that you anticipate that it is not working

Line 79. I suggest to add “for zoonotic diseases” after systems.

Methods

Lines 101-102. Please describe briefly these criteria for the reader, so he.she can assess the potential impacts on the study results

Line 119: the number of invitations sent is missing – did everyone accept to be part of the interview? if not, why?

We are also missing the distribution of participants across the 16 districts to assess the level of representativity of each district

Lines 137-138: ECoSur details organizational (at governance and operational level) and functional attributes (and not functional and governance attributes)

Line 138. The “.” should be replaced by a “;”.

Line 154 : I do not get how multiple readings ensure accuracy ; and accuracy of what? please explain

I suggest to put together paragraph lines 80-82 with paragraph lines 91-96, and move the contents of lines 83-90 to the method part.

It is usually advised that the coding is done by two people to improve accuracy. If you did not do it, you need to discuss this potential bias in the limits of the method

Lines 171-178: the contents of these 2 paragraphs should be moved to Study design and Data collection

Results

Why did you address motivation, expectations, and perceived benefits separately from factors driving and hindering collaboration?

To me they are also factors that influence collaboration. And you present them as such at the beginning of the discussion (lines 621-622).

Please clarify and explain how everything is linked, or introduce them as specific themes in part A. and merge everything together

Lines 183: why perceived? they were suggested by participants as factors? It is not what I understood in the methodology. They are perceptions of interviewees that you have identified, with your coding process, as being determinants to collaboration; but to me, they are not perceived factors per se (which suggests that interviewees identified themselves the factors)

Line 186. I do not think that an expectation can be perceived

Line 189: I do not get why assigning a unique identifier to participant will allow to link quotes to specific individuals. And I think the reader needs this information to assess the diversity of individuals you are quoting. It would be even better if you could give a code to the district where the interviewee is working to assess the representativeness of the districts

Lines 196-197: you said that the factors are grouped into 2 categories (structural and individual) but then these categories are not used in the description of the factors. In addition, in the abstract you mention sociocultural, structural and governance factors, but you do not identify these groups in the Result section. You need to harmonize across the text. Then in the discussion part, you mention again the structural and individual factors but we do not know which ones fall into these 2 categories.

Discussion

In the abstract, you mention that levers for better collaborations are stronger governance, equitable partnerships, and realistic alignment of stakeholder expectations. However, in the discussion, we do not identify these 3 main levers.

Lines 633-634. Please explain and not only make references to other articles.

Lines 655-657. You refer to participants discourse but you had references to other studies. This is inconsistent.

Lines 658-659. I do not see the link between the profit-driven dimension of collaboration and resource disparities among stakeholders. Please specify.

Lines 678-683. I do not see how ideas in this paragraph articulates with each other. How will mutual respect and engagement contribute to raise awareness and access?

It is also partly redundant with the contents of the paragraph before.

Lines 684-690 seem redundant with what has been said before (sectoral visibility)

Lines 754. You mention that focusing on Ghana prevents the generalization to other LMICs but in the abstract you state that you results are applicable to other LMICs. Please be clearer on what can be generalized and what cannot.

In addition, you have worked on a limited number of districts in a single region. Can you even generalize your results to Ghana? Please discuss this point.

Conclusions

Lines 771. This is not the first study highlighting the role of informal networks, personal initiative, and structural

Constraints in intersectoral collaboration. This has been described in some of the literature cited in this study;

Line 779. Inconsistent with the limit you have mentioned above, about the generalization of your work (see my comment above).

Line 780. Only public health outcomes? how about animal health outcomes? Economic outcomes? Social outcomes? ….

Reviewer #2: In this submission to PLOS One, the authors explore the factors influencing collaboration in zoonotic disease surveillance and response at the district (operational) level, providing insights to strengthen cross-sector collaboration and improve public health outcomes. The authors use reflexive thematic analysis to analyze interviews with 66 professionals from the human, animal, and wildlife health sectors, all directly involved in zoonotic disease surveillance and response. The authors find that their study provides insights into these realities, highlighting the need for stronger governance, equitable partnerships, and realistic alignment of stakeholder expectations.

I consider this manuscript to be of interest to machine learning researchers in biological systems as well as readers of this journal. As such, I am generally supportive of publication with a few minor comments. While the authors focus on reflexive thematic analysis, there have been recent studies using advanced machine learning methods, such as t-SNE method, which can be used efficiently for classification as well as detecting trends in biochemical systems, which should be noted: Environ. Sci. Technol. Lett. 2023, 10, 1017–1022 and Mar. Genom. 2020, 51, 100723. In particular, these prior studies showed that t-SNE can give classification of results that could help diagnose trends in biochemical systems. I am not asking the authors to carry out such t-SNE calculations, but these recent applications of t-SNE should also be noted.

.

Reviewer #1: **Yes:** Marion BordierMarion BordierMarion BordierMarion Bordier

Reviewer #2: No

---

## [Author Response · Author response to Decision Letter 1]

10 Oct 2025

Thank you very much for taking time and effort to review and improve our work.

---

## [Decision Letter · Decision Letter 1]

21 Nov 2025

Dear Dr. Dsani,

Thank you for submitting your manuscript to PLOS ONE. After careful consideration, we feel that it has merit but does not fully meet PLOS ONE’s publication criteria as it currently stands. Therefore, we invite you to submit a revised version of the manuscript that addresses the points raised during the review process.

We look forward to receiving your revised manuscript.

Kind regards,

Aravindh Babu Ramasamy Parthiban, B.V.Sc, M.V.Sc, Ph.D.

Academic Editor

PLOS ONE

Journal Requirements:

Additional Editor Comments:

Dear author,

The comments from both the reviewers requires major revision to the manuscript.

Please provide a point wise response to the comments

Thanks and regards

Reviewers' comments:

Reviewer's Responses to Questions

**Comments to the Author**

Reviewer #1: (No Response)

Reviewer #2: (No Response)

2. Is the manuscript technically sound, and do the data support the conclusions?

Reviewer #1: Partly

Reviewer #2: Partly

3. Has the statistical analysis been performed appropriately and rigorously?

Reviewer #1: N/A

Reviewer #2: Yes

4. Have the authors made all data underlying the findings in their manuscript fully available?

Reviewer #1: Yes

Reviewer #2: Yes

5. Is the manuscript presented in an intelligible fashion and written in standard English?

Reviewer #1: No

Reviewer #2: Yes

Reviewer #1: Thank you for your detailed explanations. I still have a few comments, which I outline below.

In the text below, I have kept my previous comment and your answers to them, when I was not fully satisfied, and explained why. I have also enclosed the word file which is may be easire for you to go through

Abstract

2. Comment - Line 35: when you say “governance and surveillance organization”, is

organization applied to surveillance and governance or to surveillance only? In both cases, it

sounds a bit strange to me. I wonder if “surveillance governance and organization” would

not be more suitable.

Response – Thank you for this insightful observation. We agree with your suggestion and

have revised the phrasing to ‘surveillance governance and organization systems’ in the

manuscript and subsequent figures. (Line 35)

Sorry, but it is still not clear to me. I suggested “the governance and organization of surveillance”. What do you mean by organization systems? system of what?

In addition, you announce factors influencing disease surveillance and response, but mention only surveillance here.

5. Comment - Line 45: we usually prefer to use collaboration without ‘s’

Response – Thank you, the ‘s’ has been removed. (Line 44)

But, it still remains in other parts of the manuscript

Introduction

6. Comment - Line 57: I do not see how limited public health infrastructure can favor spillover.

Can you clarify?

Response - Thank you for this important clarification. You are correct that limited public

health infrastructure does not directly favor spillover events. We have revised the sentence to

clarify that while frequent human-animal interactions and weak surveillance systems create

conditions for spillover, limited public health infrastructure facilitates the spread and impedes

effective response. (Line 56-57)

Still not clear to me. How weak surveillance systems creates conditions for spillover? The limited public health infrastructure facilitates the spread of what? I would move the weakness of surveillance to an additional factor that facilitate that impeded effective response.

Lines 75-77 soften the statement of lines 69-71 and should be put side-by-side.

Not sure that lines 71-74 are bringing relevant information in the context of this study.

Lines 77-78. Why is it specifically crucial to understand collaboration dynamics when there is a high zoonotic disease burden and under-resourced health systems? The causal link is not obvious to me

Lines 84-90 sound more like a conclusion than an introduction to the study that will be presented after. Should be rephrased.

Methods

Line 100: what do you mean by “which accounts for the higher participant numbers in one district” ?

Line 103: what do you mean by component? Before, you are talking about objectives

Lines 139-140. They agreed, but were they ultimately interviewed?

15. Comment - We are also missing the distribution of participants across the 16 districts to

assess the level of representativity of each district.

Response - Thank you for this suggestion. We understand the importance of demonstrating

representativeness. However, providing district-level distribution data would significantly

compromise participant confidentiality in our study context. The animal health sector has

very small numbers of practitioners at the district level - in many cases only 1-2 per district -

and since we interviewed complete teams in most districts, a table showing participant counts

would essentially create a map that readers familiar with the animal health sector could use to

identify specific districts. For example, if a table showed "District X has 4 animal health

participants," colleagues in the sector would immediately know which district this refers to,

as very few districts have 4 animal health officers. Combined with contextual details in

quotes, this would compromise the anonymity we promised participants.

We provide narrative information confirming that participants were recruited from across all

16 districts, with variation in numbers reflecting differences in sector sizes, purposive

sampling of key ZDSR stakeholders, and one district's involvement in another study

component (Lines 157-164). We believe this approach balances the need to demonstrate

broad geographic representation with our ethical obligation to protect participant

confidentiality in a small, highly interconnected professional community.

Sorry, but it is not clear to me how you made your sampling. On one hand you are talking about purposive and convenience sampling, and then you say that you interviewed complete teams in most districts. In addition, you talk about 19 districts first, and then you say that participants were distributed across all 16 districts Be more precise about the sampling strategy used at each stage (selection of district, selection of participants).

Lines 156-157 should be placed at the end of the previous paragraph

Lines 158-160: I do not understand the causal link in this sentence

I am not sure we need lines 160-164

Line 200: I would add discourse after participants. What do you think?

Lines 222-223 – why do you have selected no verbatims for one district?

Results

I find the text difficult to read because all of the quotes that interrupt the text. May be, you can embed them in the text. Example: Personal rapport between actors was one of the strongest enablers of collaboration. Friendly relationships fostered trust, encouraged communication, and facilitated informal information exchange. As one animal health informant explained: “It’s because we have the rapport...” (AH)

I would also suggest not to put one verbatim after each statement, but to articulate some statements together and choose the best illustrative verbatim (and the easier one to understand for people not familiar with the context, because a lot of verbatims are difficult to understand)

It would be helpful to define what an assembly coordinator is as they are mentioned in a lot of verbatims

Line 237-238. What is the difference between quotes integrated in the text and the ones provided in the supplementary file 1?

line 246: you need to better explain this quote. What is “it” referring to?

Lines 301-303 – It is not clear to me what you want to stress out. The professional recognition of who by whom create motivation of who?

Lines 304: Self-protection is illustrated later lines 312-314. So why mentioning it here?

Lines 310. What is NADMO? If we do not now, we do not see the collaboration here

Lines 341-345. I do not see how this verbatim illustrate alignment between priorities of actors. The AH actor is talking about views, not priorities. This is more in line with the fact they are both new? Cf lines 251-256

Lines 556-557. Lack of exposure to what? We need to better see the link with the theme “knowledge and training”

22. Comment - Why did you address motivation, expectations, and perceived benefits separately from factors driving and hindering collaboration?

To me they are also factors that influence collaboration. And you present them as such at the

beginning of the discussion (lines 621-622).

Please clarify and explain how everything is linked, or introduce them as specific themes in

part A. and merge everything together

Response -

Thank you for this valuable comment. We agree that motivations, expectations, and

perceived benefits all influence collaboration and should be integrated rather than treated

separately. We have thus revised the Results section to reflect this (Lines 226 - 683).

Specifically, we have:

Reorganized our results into 'Factors Influencing Intersectoral Collaboration in

ZDSR' with two categories: Individual factors (relationships, personal initiative, motivations, perceptions of superiority, shared goals), and Structural factors

(governance, visibility, workforce availability, knowledge, and resource constraints).

Positive collaborative outcomes fall within both individual and structural factors.

Integrated ‘motivations’ into the ‘individual factors’, emphasizing that they

represent participants' underlying incentives for engaging in collaboration.

Renamed ‘perceived benefits’ to ‘Positive collaborative outcomes’ and placed it as a

factor that falls under both individual and structural factors.

Separated ‘general expectations for future collaboration’ as stakeholder aspirations

to clearly distinguish aspirational views from demonstrated influences on current

collaboration.

This restructuring creates a more cohesive framework linking motivations, experiences, and expectations as interconnected influences on collaboration.

Ok, but you should mention in the introduction of the Results’ section the positive outcomes, the facts that they may be Individual of Structural factors and that’s why you address them separately

Discussion

There is still work to be done on the discussion. It could be more concise and better articulated. In addition, there are often very broad and somewhat vague recommendations, and we would like the authors to go a little further in explaining how these recommendations could be implemented in the context of Ghana and in light of their work.

Line 723. What do you mean by inequities? Is that linked to power imbalances that you are mentioning below? If yes, the link between the two paragraphs should be clearer.

I can see that you are also mentioning inequities line 732

Lines 724. I would take the opportunity of this section to discuss the fact that participants perceived that medics trained in Public health are more collaborative than the ones who are not

Line 726. I understand your point but, to me, power imbalance is real as the human health sector is usually more well-resourced than the animal health sector.

Line 737-740. But what do you propose for this to happen? We have seen that almost all African countries have now One Health Platforms but they are barely non functional and inequities as well as power imbalances persist. So how can we make OH institutionalisation working? Some studies show that too rigid institutional mechanisms also hamper collaboration because people see a risk to loose their autonomy and to be “forced” to do things they do not want to do (such as sharing data).

In addition, I am not sure that institutionalization is synonymous of equity. We have seen that institutionalization may also reinforce inequity. For instance, all international regulations which are mainly focusing on priorities of the “North” shape how funding is allocated to countries and to specific national counterparts. However those countries might have other priorities than the ones the international standards are focusing on and the funding mechanisms may create competition between national actors to accede fundings

Line 750. Poor enforcement of what?

Line 754-756. Ok, but how to achieve this? In the previous paragraph, you are calling for more institutionalisation, but now you say that it is not enough for collaboration to be effective.

You need to discuss these two points together and propose concrete solutions to that or at least way of improvement.

Line 769. I do not understand what this paragraph is adding to what is described previously. You mention physical proximity but then you are mainly refering to visibility, and are redundant with the previous paragraph. These two paragraphs should be merged and ideas better articulated

Lines 825-827. To my knowledge, it has been already addressed in the literature. See for instance: https://doi.org/10.1186/s12889-022-13878-3.

Lines 829-833. I don't see how you are proposing a roadmap; at best, you are proposing avenues for reflection and points to consider.

Lines 824- 833. I do not see how this paragraph relates to Bridging Expectation–Reality Gaps. It looks more like a summary of what you have highlighted before. It could be merged with your paragraph about the potential generalisation of your results to Ghana and LMICS and rename your sections Limits by Limits and perspectives

Line 849. Should be Third?

Lines 850-856. This is not a limitation because it was not the objective of your work to be able to generalize to Ghana and to all LMICS.

Lines 860-863. This sections could be merge with the first one to be more concise

Line 880. As stated before, I fail to see how this work has bridged the gap between policy and practice, and what you really mean with this statement. Which gap you are talking about? What is policy for you?. You have explored facilitators and barriers to collaboration at operational level, and address the importance of institutionalisation (closely linked to policy), but I don’t really see where you have improved the alignment between policy decisions and field practice.

Reviewer #2: The authors did not address my previous review comments. I have re-pasted them below. Until the authors address them, I recommend re-review of this manuscript:

I consider this manuscript to be of interest to machine learning researchers in biological systems as well as readers of this journal. As such, I am generally supportive of publication with a few minor comments. While the authors focus on reflexive thematic analysis, there have been recent studies using advanced machine learning methods, such as t-SNE method, which can be used efficiently for classification as well as detecting trends in biochemical systems, which should be noted: Environ. Sci. Technol. Lett. 2023, 10, 1017–1022 and Mar. Genom. 2020, 51, 100723. In particular, these prior studies showed that t-SNE can give classification of results that could help diagnose trends in biochemical systems. I am not asking the authors to carry out such t-SNE calculations, but these recent applications of t-SNE should also be noted.

.

Reviewer #1: **Yes:** Marion BordierMarion BordierMarion BordierMarion Bordier

Reviewer #2: No

---

## [Author Response · Author response to Decision Letter 2]

14 Jan 2026

Responses to Reviewer’s Comments (Round Two)

Reviewer 1 Comments (Dr. Marion Bordier)

We are grateful for your continued engagement with our manuscript and for your constructive feedback that has further strengthened our work.

For clarity, we have retained all comments and responses from Round 1. Our responses to your Round 2 comments follow immediately after the Round 1 responses for each point and are clearly labelled as 'Round 2'. Line numbers referenced in Round 2 responses are highlighted in yellow in the revised manuscript for easy identification.

Abstract

1. Comment (Round 1)- Line 35: when you say “governance and surveillance organization”, is organization applied to surveillance and governance or to surveillance only? In both cases, it sounds a bit strange to me. I wonder if “surveillance governance and organization” would

not be more suitable.

Response (Round 1) – Thank you for this insightful observation. We agree with your suggestion and have revised the phrasing to ‘surveillance governance and organization systems’ in the manuscript and subsequent figures. (Line 35)

Follow-up Comment (Round 2) - Sorry, but it is still not clear to me. I suggested “the governance and organization of surveillance”. What do you mean by organization systems? system of what? In addition, you announce factors influencing disease surveillance and response, but mention only surveillance here.

Response (Round 2) - Thank you for this clarification. We agree that the previous phrasing remained ambiguous. We have now revised the text to: “the governance and organization of surveillance and response systems.” This wording directly reflects the scope of the study and clearly specifies what is being governed and organized. The change has been applied in the Abstract and throughout the manuscript where relevant (Lines 35–36, 380).

2. Comment (Round 1) - Line 45: we usually prefer to use collaboration without ‘s’

Response – Thank you, the ‘s’ has been removed. (Line 44)

Follow-up Comment (Round 2) - But it still remains in other parts of the manuscript.

Response (Round 2) - Thank you for this clarification. I have reviewed the entire manuscript again to ensure consistent and intentional use of ‘collaboration’ versus ‘collaborations.’ In three sentences, I retained the plural form because the sentence refers to multiple discrete collaborative events involving different actors, which aligns with how participants described their experiences (Lines 37, 250 and 509). In all other remaining places, the term refers to collaboration as a general process, in which case the singular form is more appropriate.

These adjustments ensure clarity and accuracy while reflecting the intended meaning in each context.

Introduction

3. Comment (Round 1 - Line 57: I do not see how limited public health infrastructure can favour spill over. Can you clarify?

Response (Round 1) - Thank you for this important clarification. You are correct that limited public health infrastructure does not directly favour spill over events. We have revised the sentence to clarify that while frequent human-animal interactions and weak surveillance systems create conditions for spill over, limited public health infrastructure facilitates the spread and impedes effective response. (Line 56-57)

Follow-up Comment (Round 2) - Still not clear to me. How weak surveillance systems creates conditions for spill over? The limited public health infrastructure facilitates the spread of what? I would move the weakness of surveillance to an additional factor that facilitate that impeded effective response.

Response (Round 2) - Thank you for this important clarification. We agree with your point and have revised the sentence to clearly distinguish between factors that facilitate spill over events versus those that impede detection and response. We have now grouped weak surveillance systems together with limited public health infrastructure as factors that impede effective detection and response, rather than as conditions that create spill over.

The revised sentence now reads: “The risk of zoonotic disease transmission is particularly high in low- and middle-income countries (LMICs), where frequent human-animal interactions facilitate zoonotic spill over, while weak surveillance systems and limited public health infrastructure impede effective detection and response.” (Line 53-56)

This revision clarifies that frequent human-animal interactions are the primary driver of spill over events, while surveillance and infrastructure weaknesses affect our capacity to detect and respond to these events after they occur.

4. Comment - Lines 75-77 soften the statement of lines 69-71 and should be put side-by-side.

Not sure that lines 71-74 are bringing relevant information in the context of this study.

Response - Thank you for this helpful suggestion. We agree and have reorganized this section to improve logical flow. We have placed the two related statements about gaps in Ghana-specific research side-by-side, using the softer framing throughout. We have also removed lines 71-74 as you suggested, as they were not directly relevant to establishing the rationale for this study.

The revised paragraph now reads: “While research in other LMICs highlight the importance of institutional coordination and stakeholder engagement [13,15], empirical studies specifically examining intersectoral collaboration in Ghana's zoonotic disease surveillance and response (ZDSR) system are still emerging. Although some studies explore high-level One Health policy coordination in Ghana [16], little is known about how intersectoral collaboration occurs at the operational level within ZDSR systems” (Lines 68-73)

5. Comment Lines 77-78. Why is it specifically crucial to understand collaboration dynamics when there is a high zoonotic disease burden and under-resourced health systems? The causal link is not obvious to me.

Response - Thank you for pointing out that the causal link was not clearly articulated. We have revised this section to explicitly explain why understanding collaboration is particularly crucial in resource-constrained, high-burden settings. The revised text now clarifies that effective intersectoral collaboration is essential to maximize limited resources and prevent duplication of efforts in such contexts.

The revised text now reads: “Given Ghana's high zoonotic disease burden and under-resourced health systems [17–20], effective intersectoral collaboration is essential to make the most efficient use of limited resources and prevent duplication of efforts, making it critical to understand what facilitates or hinders collaboration in practice.” (Lines 73-76)

6. Comment: Lines 84-90 sound more like a conclusion than an introduction to the study that will be presented after. Should be rephrased.

Response - Thank you for this observation. We agree and have revised this paragraph to use more appropriate framing that introduces the study's aims rather than presenting findings.

The revised text now reads: “By examining the realities of operational-level intersectoral collaboration, this study aims to provide, evidence-based insights for developing training programs, policy adjustments, and collaboration strategies that are both effective and sustainable. Without this empirical foundation, policy initiatives alone may not translate into meaningful improvements in practice [21,22].” (Lines 80-84)

Methods

7. Comment - Line 100: what do you mean by “which accounts for the higher participant numbers in one district”?

Response - Thank you for this feedback. We agree that this information was presented prematurely at this point in the manuscript, before readers would have context about participant distribution. We have removed this phrase (from lines 91 - 92) and retained the explanation in the participant selection section (lines 159-161) where participant numbers by district are presented. Please see our response to Comment 12 for the revised explanation of participant distribution. The revised text on lines 91 – 92 now reads: “The broader project comprised multiple objectives including: (1) a rabies sectoral surveillance evaluation conducted in select districts in each area”.

8. Comment - Line 103: what do you mean by component? Before, you are talking about objectives.

Response - Thank you for catching this inconsistency in terminology. We have revised the sentence to clarify that this paper addresses one of those objectives.

The revised text now reads: “This paper uses the interview dataset from objective 2 of the study to identify specific factors influencing collaboration dynamics in the Greater Accra Metropolitan Area (GAMA).” (Line 94-97)

9. Comment - Lines 139-140. They agreed, but were they ultimately interviewed?

Response - Thank you for seeking this clarification. Yes, all three wildlife health participants who agreed to participate were ultimately interviewed. We initially did not include this detail at lines 136-137 because the final sample composition, including all three wildlife health participants and their specific roles, is presented in detail in lines 150-152. However, to provide complete transparency about the recruitment outcome and maintain parallel structure with the reporting of human and animal health sectors in this paragraph, we have added this information. The revised sentence now reads: “All three wildlife health participants contacted agreed to participate and were interviewed.” (Lines 136-137)

10. Comment (Round 1- We are also missing the distribution of participants across the 16 districts to assess the level of representativity of each district.

Response (Round 1) - Thank you for this suggestion. We understand the importance of demonstrating representativeness. However, providing district-level distribution data would significantly compromise participant confidentiality in our study context. The animal health sector has very small numbers of practitioners at the district level - in many cases only 1-2 per district - and since we interviewed complete teams in most districts, a table showing participant counts would essentially create a map that readers familiar with the animal health sector could use to identify specific districts. For example, if a table showed "District X has 4 animal health

participants," colleagues in the sector would immediately know which district this refers to, as very few districts have 4 animal health officers. Combined with contextual details in quotes, this would compromise the anonymity we promised participants. We provide narrative information confirming that participants were recruited from across all 16 districts, with variation in numbers reflecting differences in sector sizes, purposive sampling of key ZDSR stakeholders, and one district's involvement in another study component (Lines 157-164). We believe this approach balances the need to demonstrate broad geographic representation with our ethical obligation to protect participant confidentiality in a small, highly interconnected professional community.

Follow-up Comment (Round 2) - Sorry, but it is not clear to me how you made your sampling. On one hand you are talking about purposive and convenience sampling, and then you say that you interviewed complete teams in most districts. In addition, you talk about 19 districts first, and then you say that participants were distributed across all 16 districts. Be more precise about the sampling strategy used at each stage (selection of district, selection of participants).

Response (Round 2) – Thank you for this important feedback. We have substantially revised the participant selection section to provide greater clarity about our sampling strategy at each stage and to explain the progression from 21 eligible districts to the final 16 districts where interviews were conducted.

Clarification of sampling approach: We have reorganized the Methods section, moving detailed information about district selection from the Study Setting section into the Study design and participant selection section (Lines 117 – 119) where it is more appropriately positioned. We have clarified how purposive and convenience sampling were integrated. Districts and district heads were purposively selected based on their authority and direct involvement in ZDSR. However, convenience sampling elements allowed district authorities to choose whether to participate directly, nominate subordinates, or involve additional team members, provided all participants had direct ZDSR responsibilities (Lines 120 – 124).

Clarification of "complete teams": We have explained that for animal health, district teams were small (typically 1-4 officers per district), and in most cases all available animal health officers were interviewed, providing near-complete sectoral representation. For human health sectors, which have larger teams, key personnel with direct ZDSR roles were purposively selected (Lines 154 – 159).

District progression (21→19→16): We have added a detailed explanation of how the number of districts changed through the recruitment process:

• 21 districts qualified (had both human and animal health officers)

• Invitations were sent to all 21 districts, but due to incomplete or incorrect contact information, invitations reached 20 human health and 17 animal health district heads (one animal health officer oversaw two districts)

• Confirmations were received from 17 human health and 17 animal health district heads, covering 19 districts total

• Ultimately, interviews were conducted in 16 districts (human health representation in 13 districts, animal health representation in 10 districts, with only seven districts having both sectors represented)

• The reduction occurred due to data saturation, scheduling conflicts, participant availability, and time constraints (Lines 126 - 136)

These revisions provide a complete and transparent account of our sampling procedures at each stage of the study.

11. Comment - Lines 156-157 should be placed at the end of the previous paragraph

Response – Thank you for the comment. We have placed that statement at the end of the previous paragraph. (now lines 152-153)

12. Comment - Lines 158-160: I do not understand the causal link in this sentence

Response - Thank you for requesting clarification of the causal link. We have revised this section to explicitly explain why one district contributed more participants. The revised text now clarifies that this district was selected for the rabies surveillance evaluation component (mentioned earlier in lines 91-92), which required visiting all health facilities and therefore enabled broader sampling for this study while maintaining data collection efficiency.

The revised text now reads: “One district’s human health sector, however, contributed a larger number of participants (n=16) due to its involvement in the rabies surveillance evaluation of the broader research project. This evaluation required visiting all public health facilities in that district, which enabled broader sampling of ZDSR personnel beyond the typical 1-4 key stakeholders interviewed in other districts.” (Lines 159-163). This revision makes the methodological rationale clear while maintaining transparency about sampling procedures.

13. Comment - I am not sure we need lines 160-164

Response – Thank you for the comment. We agree that this information is redundant, as differences in sector sizes and the purposive sampling approach are already explained earlier in the same paragraph. We have therefore removed this sentence to avoid repetition.

14. Comment - Line 200: I would add discourse after participants. What do you think?

Response - Thank you for this suggestion. We agree that adding ‘discourse’ clarifies that themes emerged from participants' interview narratives. We have revised the sentence to read: ‘All themes reported in this manuscript were derived from multiple participants’ discourse; no single theme was based on one individual’s perspective. (Lines 199-200)

15. Comment - Lines 222-223 – why do you have selected no verbatims for one district?

Response - Thank you for seeking this clarification. Those sentences were in response to your Round 1 conc

---

## [Decision Letter · Decision Letter 2]

25 Mar 2026

Dear Dr. Dsani,

Thank you for submitting your manuscript to PLOS ONE. After careful consideration, we feel that it has merit but does not fully meet PLOS ONE’s publication criteria as it currently stands. Therefore, we invite you to submit a revised version of the manuscript that addresses the points raised during the review process.

As the corresponding author, your ORCID iD is verified in the submission system and will appear in the published article. PLOS supports the use of ORCID, and we encourage all coauthors to register for an ORCID iD and use it as well. Please encourage your coauthors to verify their ORCID iD within the submission system before final acceptance, as unverified ORCID iDs will not appear in the published article. *Only* the individual author can complete the verification step; PLOS staff  the individual author can complete the verification step; PLOS staff  the individual author can complete the verification step; PLOS staff  the individual author can complete the verification step; PLOS staff *cannot* verify ORCID iDs on behalf of authors. verify ORCID iDs on behalf of authors. verify ORCID iDs on behalf of authors. verify ORCID iDs on behalf of authors.

We look forward to receiving your revised manuscript.

Kind regards,

Julian Ruiz-Saenz, PhD

Academic Editor

PLOS One

Journal Requirements:

Additional Editor Comments:

We note that reviewer 2 has requested citation of their previous work. It is appropriate for reviewers to request that authors discuss closely related literature and cite additional sources, and in some cases the reviewer’s work may be relevant to the submitted manuscript. However, requests to cite the reviewer's work should be on topic and always be noted as optional. Here it appears that the work is outside of the scope of the current study.

PLOS ONE considers qualitative and mixed-methods studies for publication. We recommend that authors use the COREQ checklist, or other relevant checklists listed by the Equator Network, such as the SRQR, to ensure complete reporting (http://journals.plos.org/plosone/s/submission-guidelines#loc-qualitative-research). In general, we would expect qualitative studies to include the following: 1) defined objectives or research questions; 2) description of the sampling strategy, including rationale for the recruitment method, participant inclusion/exclusion criteria and the number of participants recruited; 3) detailed reporting of the data collection procedures; 4) data analysis procedures described in sufficient detail to enable replication; 5) a discussion of potential sources of bias; and 6) a discussion of limitations.

Authors conducting research in other countries or with Indigenous populations are required to complete a copy of PLOS’ questionnaire on inclusivity in global research. The policy applies to researchers who have travelled to a different country to conduct research, research with Indigenous populations or their lands, and research on cultural artefacts. You can find more information on this policy here: https://journals.plos.org/plosone/s/best-practices-in-research-reporting.

Reviewers' comments:

Reviewer's Responses to Questions

**Comments to the Author**

Reviewer #1: All comments have been addressed

Reviewer #2: (No Response)

2. Is the manuscript technically sound, and do the data support the conclusions?

Reviewer #1: Yes

Reviewer #2: (No Response)

3. Has the statistical analysis been performed appropriately and rigorously?

Reviewer #1: N/A

Reviewer #2: (No Response)

4. Have the authors made all data underlying the findings in their manuscript fully available?

The PLOS Data policy requires authors to make all data underlying the findings described in their manuscript fully available without restriction, with rare exception (please refer to the Data Availability Statement in the manuscript PDF file). The data should be provided as part of the manuscript or its supporting information, or deposited to a public repository. For example, in addition to summary statistics, the data points behind means, medians and variance measures should be available. If there are restrictions on publicly sharing data—e.g. participant privacy or use of data from a third party—those must be specified. requires authors to make all data underlying the findings described in their manuscript fully available without restriction, with rare exception (please refer to the Data Availability Statement in the manuscript PDF file). The data should be provided as part of the manuscript or its supporting information, or deposited to a public repository. For example, in addition to summary statistics, the data points behind means, medians and variance measures should be available. If there are restrictions on publicly sharing data—e.g. participant privacy or use of data from a third party—those must be specified.

Reviewer #1: Yes

Reviewer #2: (No Response)

5. Is the manuscript presented in an intelligible fashion and written in standard English?

Reviewer #1: Yes

Reviewer #2: (No Response)

Reviewer #1: Thank you for responding in such detail to all my comments.

Good luck with the finalization of this submission.

Reviewer #2:  The authors did not address my previous review comments. I have re-pasted them below. Until the authors address them, I recommend re-review of this manuscript:

I consider this manuscript to be of interest to machine learning researchers in biological systems as well as readers of this journal. As such, I am generally supportive of publication with a few minor comments. While the authors focus on reflexive thematic analysis, there have been recent studies using advanced machine learning methods, such as t-SNE method, which can be used efficiently for classification as well as detecting trends in biochemical systems, which should be noted: Environ. Sci. Technol. Lett. 2023, 10, 1017–1022 and Mar. Genom. 2020, 51, 100723. In particular, these prior studies showed that t-SNE can give classification of results that could help diagnose trends in biochemical systems. I am not asking the authors to carry out such t-SNE calculations, but these recent applications of t-SNE should also be noted.

**Do you want your identity to be public for this peer review?** For information about this choice, including consent withdrawal, please see our  For information about this choice, including consent withdrawal, please see our  For information about this choice, including consent withdrawal, please see our  For information about this choice, including consent withdrawal, please see our Privacy Policy..

Reviewer #1: **Yes:** Marion BordierMarion BordierMarion BordierMarion Bordier

Reviewer #2: No

---

## [Author Response · Author response to Decision Letter 3]

30 Mar 2026

Response to Editor – Additional Comments

Dear Editor,

We sincerely thank you for your careful evaluation of our manuscript.

1. Reviewer 2 citation requests

We appreciate your guidance regarding Reviewer 2’s request to cite their previous work. As you noted, the suggested citations appear to fall outside the scope of the present study. In line with your recommendation, we have not incorporated these citations, as they are not directly relevant to the objectives or findings of this manuscript.

2. Reporting of qualitative research (COREQ / EQUATOR guidelines)

Thank you for highlighting the importance of comprehensive reporting for qualitative and mixed-methods research.

In response:

- We have completed and submitted the COREQ checklist.

- We confirm that all key elements recommended for qualitative reporting—including research objectives, sampling strategy, data collection procedures, analytical approach, reflexivity, and study limitations—are fully addressed within the manuscript and/or detailed in the COREQ checklist.

As such, no further changes to the manuscript were required, as the submitted version already complies with these reporting standards. The COREQ checklist serves to clarify and transparently indicate where this information is presented.

3. Inclusivity in global research

We appreciate the opportunity to address the journal’s policy on inclusivity in global research.

In response:

- We have carefully cross-checked and resubmitted the Inclusivity in Global Research questionnaire as a Supporting Information file to ensure that all referenced page numbers accurately correspond to the current version of the manuscript.

- The manuscript already includes the relevant ethical, cultural, and collaborative considerations, and references to this checklist are clearly indicated and/or further clarified where appropriate.

4. Summary

We are grateful for your guidance, which has helped ensure that our manuscript aligns fully with the journal’s reporting and policy requirements. We believe that the submitted materials now comprehensively address all points raised.

We thank you for your time and consideration and hope the manuscript is now suitable for publication in PLOS ONE.

Kind regards,

Joannishka Dsani, DVM, MA

PhD Candidate

Centre for Development Research,

University of Bonn, Germany

---

## [Editor Report · Decision Letter 3]

3 Apr 2026

Beyond silos: Drivers and barriers to intersectoral collaboration in zoonotic disease surveillance and response in the Greater Accra Metropolitan Area, Ghana

PONE-D-25-16092R3

Dear Dr. Dsani,

We’re pleased to inform you that your manuscript has been judged scientifically suitable for publication and will be formally accepted for publication once it meets all outstanding technical requirements.

Kind regards,

Julian Ruiz-Saenz, PhD.

Academic Editor

PLOS One
---

## [Editor Report · Acceptance letter]

PONE-D-25-16092R3

PLOS One

Dear Dr. Dsani,

I'm pleased to inform you that your manuscript has been deemed suitable for publication in PLOS One. Congratulations! Your manuscript is now being handed over to our production team.

Kind regards,

on behalf of

Dr. Julian Ruiz-Saenz

Academic Editor

PLOS One